# Hyperbolic-PDE GNN: Spectral Graph Neural Networks in the Perspective of A System of Hyperbolic Partial Differential Equations

**Juwei Yue** [1] [2]   **Haikuo Li** [1] [2]   **Jiawei Sheng** [1] [2]   **Xiaodong Li** [1] [2]   **Taoyu Su** [1] [2]   **Tingwen Liu** [1] [2]   **Li Guo** [1] [2]

## Abstract

Graph neural networks (GNNs) leverage message passing mechanisms to learn the topological features of graph data. Traditional GNNs learns node features in a spatial domain unrelated to the topology, which can hardly ensure topological features. In this paper, we formulates message passing as a system of hyperbolic partial differential equations (hyperbolic PDEs), constituting a dynamical system that explicitly maps node representations into a particular solution space. This solution space is spanned by a set of eigenvectors describing the topological structure of graphs. Within this system, for any moment in time, a node features can be decomposed into a superposition of the basis of eigenvectors. This not only enhances the interpretability of message passing but also enables the explicit extraction of fundamental characteristics about the topological structure. Furthermore, by solving this system of hyperbolic partial differential equations, we establish a connection with spectral graph neural networks (spectral GNNs), serving as a message passing enhancement paradigm for spectral GNNs. We further introduce polynomials to approximate arbitrary filter functions. Extensive experiments demonstrate that the paradigm of hyperbolic PDEs not only exhibits strong flexibility but also significantly enhances the performance of various spectral GNNs across diverse graph tasks. Our code is released at https://github.com/YueAWu/Hyperbolic-GNN.

---

[1]Institute of Information Engineering, Chinese Academy of Sciences, Beijing, China [2]School of Cyber Security, University of Chinese Academy of Sciences, Beijing, China. Correspondence to: Jiawei Sheng <shengjiawei@iie.ac.cn>.

*Proceedings of the $42^{nd}$ International Conference on Machine Learning*, Vancouver, Canada. PMLR 267, 2025. Copyright 2025 by the author(s).

*Table 1.* Comparison between traditional MP-based (Line 1) and hyperbolic PDE-based (Line 2) paradigm, where $\mathbf{e}$ is the unit vector, $\lambda$ and $\mathbf{u}$ are the eigenvalue and eigenvector of Laplacian.

| | Paradigm | Bases of Solution Space |
|---|---|---|
| (1) | $\mathbf{h}_i^{(l+1)} = \mathcal{U}_i\left(\mathbf{h}_i^{(l)}, \mathcal{A}_j\left(\mathcal{M}_{ij}\left(\mathbf{h}_i^{(l)}, \mathbf{h}_j^{(l)}, \mathbf{e}_{ij}\right)\right)\right)$ | $\mathbf{e}_1, \mathbf{e}_2, \ldots, \mathbf{e}_d$ |
| (2) | $\frac{\partial^2 \mathbf{X}}{\partial t^2} = a^2 \hat{\mathbf{L}} \mathbf{X} \rightarrow \frac{d\mathbf{w}_l}{dt} = \begin{bmatrix} \mathbf{I}_n & \mathbf{0} \\ \mathbf{0} & a^2\hat{\mathbf{L}} \end{bmatrix} \mathbf{w}_l$ | $e^{\lambda_1' t}\mathbf{u}_1', e^{\lambda_2' t}\mathbf{u}_2', \ldots, e^{\lambda_{2n}' t}\mathbf{u}_{2n}'$ |

## 1. Introduction

Currently, graph neural networks (GNNs) ([Wu et al., 2021](#)) have experienced rapid development due to the widespread usage of graph data. They have not only achieved significant advancements in artificial intelligence fields such as recommendation systems ([Wu et al., 2023](#); [Liao et al., 2025](#)), social networks ([Lin et al., 2021](#); [Zhu et al., 2023](#)), and anomaly detection ([Ma et al., 2023](#); [Gao et al., 2023](#)), but have also promoted progress in various other domains such as particle physics ([Shlomi et al., 2021](#)) and drug discovery ([Gaudelet et al., 2021](#)).

Mainstream methods are spectral GNNs represented by GCN ([Kipf & Welling, 2017](#)). The key idea of these methods is to utilize message passing (MP) ([Gilmer et al., 2017](#)) to explore the topological features of the graphs. Specifically, MP aggregates information from neighbors based on the topology of graphs. The essence of MP involves mapping node features into the spatial domain to the spectral domain through Fourier transformation, performing graph convolutions in the spectral domain, and then inversely transforming them back into the spatial domain. However, the issue lies in the fact that, even if node features learn the topology of the graph, they are still mapped back to a spatial domain unrelated to the topology. As shown in Table 1, traditional MP-based GNNs obtain node feature vectors that are actually situated in the Euclidean space with unit vector bases, which can hardly ensure topological feature therein, potentially learning redundant information. (In contrast, our hyperbolic PDE-based model derives node feature vectors with bases stemmed from Laplacian, incorporating the underlying topological information. Detailed later.)

Current researches ([Rusch et al., 2022](#); [Chamberlain et al., 2021](#); [Li et al., 2024](#)) model MP using differential equa-

tions (DEs), formulating MP as a dynamical system that captures the evolution of messages over time. It provides an interpretable way to embed node feature vectors into spaces with particular properties. We utilize the solution of DEs as the node embedding space, and propose a graph neural network named **Hyperbolic-PDE GNN** based on *a system of hyperbolic partial differential equations (PDEs)*. We theoretically demonstrate that by modeling MP as the paradigm of the system of hyperbolic PDEs, the solution space (*i.e.*, the space where node features reside) is spanned by a set of orthogonal bases describing topological features, with these bases fundamentally stemming from the eigenvectors of the Laplacian matrix. This proof of the solution space is derived by converting the system of hyperbolic PDEs into a system of first-order constant-coefficient homogeneous linear differential equations. Therefore, the node features obtained from this paradigm can be expanded into a linear combination of these eigenvector basis, where node features inherently contain the fundamental characteristics describing the topological structure, and messages are propagated along the directions of the eigenvectors. Benefiting from the property, our Hyperbolic-PDE GNN enjoys an advantage in generating better node features with topology of graphs.

Lastly, we discover that solving this system of equations requires eigen decomposition, which incurs significant computational costs in practice. Besides, the original system of equations with vanilla Laplacian struggles to model complex nonlinear relationships, resulting in unsatisfactory model flexibility and robustness. Therefore, we further introduce polynomials to approximate the solution space. Some GNNs based on polynomial spectral filters (He et al., 2021; Wang & Zhang, 2022; He et al., 2022) have demonstrated the advantage of polynomials in approximating arbitrary filter functions. Leveraging this excellent property, we can ensure that the approximate solution space does not deviate too far from the original solution space and can describe more complex nonlinear relationships, greatly enhancing the flexibility of model. Furthermore, owing to the utilization of polynomials, we establish a connection between Hyperbolic-PDE GNNs and traditional spectral GNNs, which can further enhance the performance of these methods. In summary, the contributions can be summarized as follows:

- We propose a novel Hyperbolic-PDE GNN based on a system of hyperbolic PDEs, and theoretically demonstrate that node features encapsulate fundamental characteristics of the topological structure.

- By utilizing polynomials to the solution space, we establish connections between Hyperbolic-PDE GNNs and traditional GNNs to further improve performance.

- Extensive experiments indicate that Hyperbolic-PDE GNN is effective and the paradigm of the system of

hyperbolic partial differential equations can enhance the performance of traditional GNNs.

## 2. Preliminaries

### 2.1. Notations of Graph

A simple undirected graph $\mathcal{G} = (\mathcal{V}, \mathcal{E})$ consists of a set of nodes $\mathcal{V}$ and a set of edges $\mathcal{E}$. A feature matrix $\mathbf{X} \in \mathbb{R}^{n \times d}$ and a adjacency matrix $\mathbf{A}$ represent respectively the features and structure of the graph, where $n = |\mathcal{V}|$ is the number of nodes and $d$ is the dimension of features. In the graph signal processing (GSP) (Shuman et al., 2013), the symmetric normalized Laplacian is $\mathbf{L} = \mathbf{I} - \mathbf{D}^{-1/2}\mathbf{A}\mathbf{D}^{-1/2}$, where $\mathbf{D}$ denotes the diagonal degree matrix of $\mathbf{A}$. The Laplacian can be decomposed into $\mathbf{L} = \mathbf{U}\mathbf{\Lambda}\mathbf{U}^\top$, where $\mathbf{\Lambda} = \mathrm{diag}\{\lambda_1, \lambda_2, \ldots, \lambda_n\}$ is a diagonal matrix of eigenvalues and $\mathbf{U} = [\mathbf{u}_1, \mathbf{u}_2, \ldots, \mathbf{u}_n] \in \mathbb{R}^{n \times n}$ is the matrix of corresponding eigenvectors.

### 2.2. Spectral Graph Neural Networks

Spectral Graph Neural Networks achieve *Spectral Graph Convolution*

$$\mathbf{Z} = \mathbf{U}\mathbf{g}(\Lambda)\mathbf{U}^\top\mathbf{X} \tag{1}$$

by designing *Graph Spectral Filter* $\mathbf{g}(\Lambda)$, where $\mathbf{g}(\Lambda) = \mathrm{diag}\{g(\lambda_1), \ldots, g(\lambda_n)\}$. To fit an ideal filter, $g(\lambda)$ is often approximated using a $K$-th order polynomial:

$$g(\lambda) := \sum_{k=0}^{K} \theta_k \lambda^k, \tag{2}$$

where $\theta_k$ is the filter coefficient. Then, the Spectral Graph Convolution is given by:

$$\mathbf{Z} = \sum_{k=0}^{K} \theta_k \mathbf{U}\Lambda^k\mathbf{U}^\top\mathbf{X} = \sum_{k=0}^{K} \theta_k \mathbf{L}^k\mathbf{X}. \tag{3}$$

Table 2 presents some classic examples of spectral GNNs.

### 2.3. Hyperbolic Partial Differential Equation

A hyperbolic partial differential equation is often used in the real world to describe phenomena involving vibrations: if a point in space is disturbed by an initial value, this disturbance will propagate through space at a finite speed, affecting certain points in space. The propagation pattern of the disturbance is wave-like and propagates along the characteristic directions of the equation. The most common form of hyperbolic partial differential equations is given by:

$$\frac{\partial^2 u}{\partial t^2} = a^2 \sum_{l=1}^{d} \frac{\partial^2 u}{\partial \omega_l^2}, \tag{4}$$

*Table 2.* Spectral Graph Neural Networks.

| Methods (Polynomial Basis) | Graph Spectral Filter | Spectral Graph Convolution |
|---|---|---|
| GCN (Monomial) | $g(\widetilde{\lambda}) = (1 - \widetilde{\lambda}), \ \widetilde{\lambda} \in [0, 2)$ | $\mathbf{Z} = (\mathbf{I} - \widetilde{\mathbf{L}})\mathbf{X}\mathbf{W}$ |
| SGC (Monomial) | $g(\widetilde{\lambda}) = (1 - \widetilde{\lambda})^K, \ \widetilde{\lambda} \in [0, 2)$ | $\mathbf{Z} = (\mathbf{I} - \widetilde{\mathbf{L}})^K \mathbf{X}\mathbf{W}$ |
| APPNP (Monomial) | $g(\widetilde{\lambda}) = \sum_{k=0}^{K} \frac{\alpha^k}{1-\alpha}(1 - \widetilde{\lambda})^k, \ \widetilde{\lambda} \in [0, 2)$ | $\mathbf{Z} = \alpha(\mathbf{I} - (1 - \alpha)(\mathbf{I} - \widetilde{\mathbf{L}}))^{-1}\phi(\mathbf{X})$ |
| GPR-GNN (Monomial) | $g(\widetilde{\lambda}) = \sum_{k=0}^{K} \theta_k (1 - \widetilde{\lambda})^k, \ \widetilde{\lambda} \in [0, 2)$ | $\mathbf{Z} = \sum_{k=0}^{K} \theta_k (\mathbf{I} - \widetilde{\mathbf{L}})^k \phi(\mathbf{X})$ |
| ChebNet (Chebyshev Polynomial) | $g(\lambda) = \sum_{k=0}^{K-1} \theta_k T_k(\lambda - 1), \ \lambda \in [0, 2]$ | $\mathbf{Z} = \sum_{k=0}^{K-1} T_k(\mathbf{L} - \mathbf{I})\mathbf{X}\mathbf{W}_k$ |
| BernNet (Bernstein Polynomial) | $g(\lambda) = \sum_{k=0}^{K} \theta_k b_{k,K}(\lambda/2), \ \lambda \in [0, 2]$ | $\mathbf{Z} = \sum_{k=0}^{K} \frac{\theta_k}{2^K} \binom{K}{k}(2\mathbf{I} - \mathbf{L})^{K-k}\mathbf{L}^k \phi(\mathbf{X})$ |
| JacobiNet (Jacobi Polynomial) | $g(\lambda) = \sum_{k=0}^{K} \theta_k P_k^{a,b}(1 - \lambda), \ \lambda \in [0, 2]$ | $\mathbf{Z}_{:j} = \sum_{i=0}^{K} \theta_{kj} P_k^{a,b}(\mathbf{I} - \mathbf{L})\phi(\mathbf{X}_{:j})$ |
| ChebNetII (Chebyshev Polynomial) | $g(\lambda) = \sum_{k=0}^{K} c_k(\theta) T_k(\lambda - 1), \ \lambda \in [0, 2]$ | $\mathbf{Z} = \frac{2}{K+1} \sum_{k=0}^{K} \sum_{j=0}^{K} \theta_j T_k(x_j) T_k(\mathbf{L} - \mathbf{I})\phi(\mathbf{X})$ |

where $u(\omega_1, \omega_2, \ldots, \omega_d, t)$ represents the amplitude, $t \in [0, \infty)$ denotes time, $(\omega_1, \omega_2, \ldots, \omega_d)$ stands for the coordinates in a $d$-dimensional space $\Omega^d$, and $a$ signifies the propagation velocity.

## 3. Methodology

### 3.1. A System of Hyperbolic PDEs on Graph

Considering the right-hand side of Equation (4), it is essentially expressed by a Laplacian operator $\Delta = \nabla \cdot \nabla$, which can then be written in the form of the divergence of the gradient of $u$:

$$\frac{\partial^2 u}{\partial t^2} = a^2 \nabla \cdot \nabla u, \tag{5}$$

where $\nabla$ and $\nabla \cdot$ denote the operator of gradient and divergence, respectively. Here, the gradient and divergence are defined on a manifold. In graphs, nodes are a set of discrete points, thus we first discretize the amplitude $u$ on the manifold into features $\mathbf{X}(t)$ in the spatial domain, where each element $x_{il} = x_{il}(t)$ denotes the feature of each node $v_i$ on each dimensions $l = 1, 2, \ldots, d$ at time $t$. Next, we express the gradient on the graph as the feature difference on any dimension $l$ between a node $v_i$ and its neighbors $v_j \in \mathcal{N}(v_i)$:

$$\nabla x_{il} := x_{il} - x_{jl}, \tag{6}$$

with the direction pointing from the neighbor towards the node, indicating the direction of information flow. Finally, we define the divergence on the graph as the total feature difference on any dimension between a node and all its neighbors:

$$\nabla \cdot \nabla x_{il} = \sum_{v_j \in \mathcal{N}(v_i)} (x_{il} - x_{jl}), \tag{7}$$

since divergence essentially involves a summation operation. Therefore, for a node $v_i$ on any dimension $l$, its hyperbolic partial differential equation is:

$$\frac{\partial^2 x_{il}}{\partial t^2} = a^2 \sum_{v_j \in \mathcal{N}(v_i)} (x_{il} - x_{jl}). \tag{8}$$

When we combine the hyperbolic partial differential equations of all nodes on all dimensions, we obtain a system of hyperbolic partial differential equations. For simplicity, we can rewrite it in matrix form:

$$\frac{\partial^2 \mathbf{X}}{\partial t^2} = a^2 \widehat{\mathbf{L}} \mathbf{X}. \tag{9}$$

To distinguish from matrix $\mathbf{L}$, hereafter we will denote the eigenvalues and eigenvectors of matrix $\widehat{\mathbf{L}}$ as $\widehat{\lambda}$ and $\widehat{\mathbf{u}}$. *Detailed derivation of the system of partial differential equations can be found in Appendix A.1.*

### 3.2. Derivation of Solution Space

For the system of partial differential equations (9), we require the existence of solutions for this system to ensure the meaningful extraction of node features. Next, we will introduce the theorem regarding the existence of solutions for this system of partial differential equations.

**Theorem 3.1.** *For given system of differential equations (9) and a graph signal on any dimension $l$, there exists a **solution space** determined by a **fundamental matrix of solution**:*

$$\mathbf{\Phi}(t) = \begin{bmatrix} \exp \mathbf{I}t & \mathbf{0} \\ \mathbf{0} & a^2 \exp \widehat{\mathbf{L}}t \end{bmatrix}. \tag{10}$$

*Therefore, a graph signal on any dimension $l$ can be expressed as a linear combination of $2n$ linearly independent column vectors $\varphi_1(t), \varphi_2(t), \ldots, \varphi_{2n}(t)$ from the fundamental matrix of solution $\mathbf{\Phi}(t)$.*

By performing a variable substitution on Equation (9), we transform it into a system of first-order constant-coefficient homogeneous linear differential equations, and successfully complete the proof using relevant theory. *See Appendix A.2 for proof.*

*Remark* 3.2. Theorem 3.1 ensures the existence of a solution when modeling message passing as a system of hyperbolic partial differential equation. Here, the solution corresponds to the node features. In other words, **we can definitely obtain precise node representation through Equation (9)**.

Although we obtain a fundamental matrix of solution of Equation (9), in order to determine it, we need to calculate $\exp \widehat{\mathbf{L}} t$. However, $\exp \widehat{\mathbf{L}} t$ is defined by a matrix series, it is generally difficult to obtain directly. Therefore, we introduce the following theorem to indirectly find the fundamental matrix of solution.

**Theorem 3.3.** *For given system of differential equations (9) and a graph signal on any dimension $l$, matrix*

$$\mathbf{C} = \begin{bmatrix} \mathbf{I} & \mathbf{0} \\ \mathbf{0} & a^2\widehat{\mathbf{L}} \end{bmatrix} \tag{11}$$

*has eigenvalues $\lambda_1' = \lambda_2' = \cdots = \lambda_n' = 1, \lambda_{n+1}' = a^2\widehat{\lambda}_1, \lambda_{n+2}' = a^2\widehat{\lambda}_2, \ldots, \lambda_{2n}' = a^2\widehat{\lambda}_n$ and corresponding $2n$ linearly independent eigenvectors*

$$\mathbf{u}_1' = \begin{bmatrix} \mathbf{v}_1 \\ \mathbf{0} \end{bmatrix}, \mathbf{u}_2' = \begin{bmatrix} \mathbf{v}_2 \\ \mathbf{0} \end{bmatrix}, \ldots, \mathbf{u}_n' = \begin{bmatrix} \mathbf{v}_n \\ \mathbf{0} \end{bmatrix},$$
$$\mathbf{u}_{n+1}' = \begin{bmatrix} \mathbf{0} \\ \widehat{\mathbf{u}}_1 \end{bmatrix}, \mathbf{u}_{n+2}' = \begin{bmatrix} \mathbf{0} \\ \widehat{\mathbf{u}}_2 \end{bmatrix}, \ldots, \mathbf{u}_{2n}' = \begin{bmatrix} \mathbf{0} \\ \widehat{\mathbf{u}}_n \end{bmatrix}, \tag{12}$$

*where $\mathbf{v}_1 = [1, 0, \ldots, 0]^\top, \mathbf{v}_2 = [0, 1, \ldots, 0]^\top, \ldots, \mathbf{v}_n = [0, 0, \ldots, 1]^\top \in \mathbb{R}^n$ are corresponding eigenvectors of $\mathbf{I}$, then matrix*

$$\mathbf{\Phi}(t) = [e^{\lambda_1' t}\mathbf{u}_1', e^{\lambda_2' t}\mathbf{u}_2', \ldots, e^{\lambda_{2n}' t}\mathbf{u}_{2n}'] \tag{13}$$

*is a fundamental matrix of solution of Equation (9). Particularly, $\exp \mathbf{C} t = \mathbf{\Phi}(t)\mathbf{\Phi}^{-1}(0)$.*

*See Appendix A.3 for proof.*

*Remark* 3.4. Theorem 3.3 reveals that the solutions of Equation (9) lie within a particular space, which is spanned by a basis of eigenvectors describing the topological structure of the graph. In other words, **in the modeling of a system of hyperbolic partial differential equations, the node features propagate messages along specific directions of eigenvectors**. The theorem not only enhances the interpretability of the paradigm of a system of hyperbolic partial differential equation but also ensures that the node features imply the fundamental characteristics of the topological structure of the graph.

### 3.3. Polynomial Approximation of Solution

Note that Theorem 3.1 and Theorem 3.3 explicitly expound that for any graph signal in Equation (9), there exists a solution space determined by the topological structure of the graph and the propagation coefficient $a$. Among these, $\mathbf{L}$ is a constant matrix, thus the only parameter that can be uniquely determined by the model is the propagation coefficient $a$. If we solely rely on $a$ to regulate message passing, it is not only hard to depict intricate nonlinear relationships but also inevitably leads to models with *significantly reduced flexibility and robustness*. Moreover, the fundamental

matrix of solution of Equation (13) still requires the computation of the eigenvalues and eigenvectors of the Laplacian matrix. Its computational complexity is $O(n^3)$, which can *significantly impact the efficiency of the model* when dealing with large-scale graphs. Therefore, in this section, we introduce polynomial approximation theory (Stone, 1937) to address the aforementioned issue.

**Theorem 3.5.** *For a function $f(x)$ that is continuous on $[a, b]$, for any $\varepsilon > 0$, there always exists an algebraic polynomial $P(x)$ such that*

$$\|f(x) - P(x)\|_\infty < \varepsilon \tag{14}$$

*holds uniformly on $[a, b]$.*

The Bernstein polynomials have been demonstrated to hold uniformly on $[0, 1]$ and have been practically applied in He et al. (2021). Orthogonal polynomials are crucial tools for function approximation, where any polynomial can be expressed as a linear combination of orthogonal polynomials. Therefore, we approximate the right-hand side of Equation (9) using a polynomial with respect to Laplacian:

$$\frac{\partial^2 \mathbf{X}}{\partial t^2} = a^2\widehat{\mathbf{L}}\mathbf{X} \approx P(\mathbf{L}, t)\mathbf{X} = \sum_{k=0}^{K} \theta_{tk} p_k(\mathbf{L})\mathbf{X}, \tag{15}$$

where $p_k(\cdot)$ is the $k$-th order orthogonal polynomial. Here, we replace the basic Laplacian matrix $\widehat{\mathbf{L}}$ with the more commonly used symmetric normalized Laplacian matrix $\mathbf{L}$, which differs only in the nonzero elements.

As introduced in Section Preliminaries, it is due to the achievements of pioneers in the field of graphs who introduced Laplacian polynomials into graph data that we have laid the foundation for establishing the connection between the system of hyperbolic partial differential equations and spectral GNNs. As shown in Table 2, Chebyshev polynomials serve as an example of utilizing orthogonal polynomials to approximate graph spectral filter. The distinction lies in the fact that in this paper, we employ polynomial to approximate the solution space of the system of hyperbolic partial differential equations. Then, based on Chebyshev polynomials, Equation (15) can be rewritten as:

$$\frac{\partial^2 \mathbf{X}}{\partial t^2} = \sum_{k=0}^{K-1} T_k(\mathbf{L} - \mathbf{I})\mathbf{X}\mathbf{W}_k. \tag{16}$$

### 3.4. Hyperbolic-PDE GNNs

For a given PDE, solving an analytical solution is often challenging, numerical methods are commonly employed to approximate its solution in mathematics. This technique is currently extensively utilized within the realm of graph studies, with the **forward Euler method** standing out as the predominant choice (Eliasof et al., 2021; Chamberlain

*Table 3.* Dataset statistics.

| Datesets | #Nodes | #Edges | #Features | #Classes |
|---|---|---|---|---|
| Cora | 2708 | 5278 | 1433 | 7 |
| CiteSeer | 3327 | 4552 | 3703 | 6 |
| PubMed | 19717 | 44324 | 500 | 3 |
| Computers | 13752 | 245861 | 767 | 10 |
| Photo | 7650 | 119081 | 745 | 8 |
| CS | 18333 | 81894 | 6805 | 15 |
| Actor | 7600 | 30019 | 932 | 5 |
| Texas | 183 | 325 | 1703 | 5 |
| Cornell | 183 | 298 | 1703 | 5 |
| DeezerEurope | 28281 | 92752 | 128 | 2 |

et al., 2021; Thorpe et al., 2022; Li et al., 2024; Yue et al., 2025). Specifically, the forward Euler method discretizes continuous time, dividing time into $m$ equidistant segments with a time step $\tau$: $t = t_m = m\tau, m = 0, 1, \ldots$. The initial and terminal time are denoted as $t_0$ and $T = t_T$, and $t_m$ represents the $m$-th time moment in time.

After discretizing time, Equation (15) can be reformulated into the following difference scheme:

$$\frac{\dot{\mathbf{X}}(t_m) - \dot{\mathbf{X}}(t_{m-1})}{\tau} = P(\mathbf{L}, t_m)\mathbf{X}(t_m), \qquad (17)$$

where $\dot{\mathbf{X}}(t_m)$ and $\dot{\mathbf{X}}(t_{m-1})$ are the first derivative w.r.t time: $\dot{\mathbf{X}}(t_m) = \frac{\mathbf{X}(t_{m+1}) - \mathbf{X}(t_m)}{\tau}, \dot{\mathbf{X}}(t_{m-1}) = \frac{\mathbf{X}(t_m) - \mathbf{X}(t_{m-1})}{\tau}$. Furthermore, to solve Equation (17), we initialize the initial values of node features and its first derivatives as:

$$\mathbf{X}(t_0) = \phi_0(\mathbf{X}), \dot{\mathbf{X}}(t_0) = \frac{\mathbf{X}(t_1) - \mathbf{X}(t_{-1})}{2\tau} = \phi_1(\mathbf{X}), \quad (18)$$

where $\phi_0(\cdot)$ and $\phi_1(\cdot)$ can be obtained from nonlinear layers. Substituting Equation (18) into Equation (17), we obtain the initial value at 1st time step as:

$$\mathbf{X}(t_1) = \tau\dot{\mathbf{X}}(t_0) + \left(\mathbf{I} + \frac{\tau^2}{2}P(\mathbf{L}, t_0)\right)\mathbf{X}(t_0). \qquad (19)$$

Given $\mathbf{X}(t_0)$ and $\mathbf{X}(t_1)$, we can derive the updating iterative formula for the node features at $m$-th time step:

$$\mathbf{X}(t_{m+1}) = \left(2\mathbf{I} + \tau^2 P(\mathbf{L}, t_m)\right)\mathbf{X}(t_m) - \mathbf{X}(t_{m-1}). \quad (20)$$

# 4. Experiments

## 4.1. Node Classification on Graphs

We evaluate the performance of Hyperbolic-PDE GNN on the classic node classification task (Shchur et al., 2018). Following the practice of Wang & Zhang (2022), we conduct experiments on following real-world datasets: a) homophilic datasets include three citation networks, Cora, CiteSeer, and PubMed (Shchur et al., 2018), two Amazon co-purchase

*Table 4.* The results of spectral GNNs: mean accuracy (%)± standard deviation on 60%/20%/20% data splits on 10 runs. **Bold** and underline indicate optimal and suboptimal results.

| Methods | Cora | CiteSeer | PubMed | Actor |
|---|---|---|---|---|
| GCN | $87.14_{\pm1.01}$ | $79.86_{\pm0.67}$ | $86.74_{\pm0.27}$ | $33.23_{\pm1.16}$ |
| APPNP | $88.14_{\pm0.73}$ | $80.47_{\pm0.74}$ | $88.12_{\pm0.31}$ | $39.66_{\pm0.55}$ |
| ARMA | $87.13_{\pm0.80}$ | $80.04_{\pm0.55}$ | $86.93_{\pm0.24}$ | $37.67_{\pm0.54}$ |
| GPR-GNN | $88.57_{\pm0.69}$ | $80.12_{\pm0.83}$ | $88.46_{\pm0.33}$ | $39.92_{\pm0.67}$ |
| ChebNet | $86.67_{\pm0.82}$ | $79.11_{\pm0.75}$ | $87.95_{\pm0.28}$ | $37.61_{\pm0.85}$ |
| BernNet | $88.95_{\pm0.95}$ | $80.09_{\pm0.97}$ | $88.48_{\pm0.41}$ | $41.79_{\pm1.01}$ |
| JacobiConv | $88.98_{\pm0.46}$ | $80.78_{\pm0.79}$ | $89.62_{\pm0.41}$ | $41.17_{\pm0.64}$ |
| ChebNetII | $88.71_{\pm0.93}$ | $80.53_{\pm0.79}$ | $88.93_{\pm0.29}$ | $41.75_{\pm1.07}$ |
| EvenNet | $87.77_{\pm0.67}$ | $78.51_{\pm0.63}$ | $90.87_{\pm0.34}$ | $40.36_{\pm0.65}$ |
| OptBasisGNN | $87.00_{\pm1.55}$ | $80.58_{\pm0.82}$ | $90.30_{\pm0.19}$ | $\mathbf{42.39_{\pm0.52}}$ |
| PyGNN | $88.34_{\pm0.31}$ | $79.49_{\pm0.45}$ | $89.52_{\pm0.24}$ | – |
| SpecFormer | $88.57_{\pm1.01}$ | $\underline{81.49_{\pm1.32}}$ | $87.73_{\pm0.58}$ | $41.93_{\pm1.04}$ |
| NFGNN | $\underline{89.82_{\pm0.43}}$ | $80.56_{\pm0.55}$ | $89.89_{\pm0.68}$ | $40.62_{\pm0.38}$ |
| UniFilter | $89.49_{\pm1.35}$ | $81.39_{\pm1.32}$ | $\mathbf{91.44_{\pm0.50}}$ | $40.84_{\pm1.21}$ |
| **Our methods** | $\mathbf{90.82_{\pm0.95}}$ | $\mathbf{81.88_{\pm1.33}}$ | $\underline{91.36_{\pm0.59}}$ | $\underline{42.03_{\pm1.59}}$ |

network, Computers and Photo (Yang et al., 2016), and co-author network CS (Shchur et al., 2018); b) heterophilic datasets include co-occurrence graph Actor, two WebKB datasets, Texas and Cornell (Pei et al., 2020), and social network DeezerEurope (Rozemberczki & Sarkar, 2020). Note that we do not utilize the Chameleon and Squirrel datasets (Rozemberczki et al., 2019) because they have serious issues of node duplication (Platonov et al., 2023). Table 3 shows the dataset statistics in our paper.

We choose classical spectral GNNs as baselines: GCN (Kipf & Welling, 2017), SGC (Wu et al., 2019), APPNP (Klicpera et al., 2019a), ARMA (Bianchi et al., 2021), GPR-GNN (Chien et al., 2021), ChebNet (Defferrard et al., 2016), BernNet (He et al., 2021), EvenNet (Lei et al., 2022), JacobiConv (Wang & Zhang, 2022), ChebNetII (He et al., 2022), OptBasisGNN (Guo & Wei, 2023), PyGNN (Geng et al., 2023), SpecFormer (Bo et al., 2023), NFGNN (Zheng et al., 2024), and UniFilter (Huang et al., 2024b).

We split the datasets into training/validation/test sets in 60%/20%/20% and report the mean accuracy and standard deviation of the models over 10 random splits. For our method, we utilize cross-entropy as the loss function and optimize parameters using the Adam optimizer (Kingma & Ba, 2015). We conduct experiments on a NVIDIA Tesla V100 GPU with 16GB memory and Intel Xeon E5-2660 v4 CPUs. The experimental code is implemented based on Py-Torch and PyTorch Geometric (Fey & Lenssen, 2019), and we employ wandb [1] for parameter search. *See Appendix C for specific experimental settings.*

Table 4 illustrates the performance of Hyperbolic-PDE GNN on four prominent benchmarks. It can be observed that Hyperbolic-PDE GNN achieves optimal performance on Cora and CiteSeer, with a 1% improvement on Cora. Ad-

---

[1] https://github.com/wandb/wandb

*Table 5.* The results of base models and Hyperbolic-PDE GNN: mean accuracy (%)±standard deviation on 60%/20%/20% data splits on 10 runs. **Bold** indicates optimal results for all methods, and **Bold** indicates optimal results between Hyperbolic-PDE GNN and base model. ∗ indicates reproduced results.

| Methods | Cora | CiteSeer | PubMed | Computers | Photo | CS* | Actor | Texas | Cornell | DeezerEurope* |
|---|---|---|---|---|---|---|---|---|---|---|
| SGC | $85.48_{\pm1.48}$ | $\mathbf{80.75_{\pm1.15}}$ | $85.36_{\pm0.52}$ | $\mathbf{88.19_{\pm0.45}}$* | $93.60_{\pm0.90}$* | $95.13_{\pm0.25}$ | $28.81_{\pm1.11}$ | $81.31_{\pm3.30}$ | $72.62_{\pm9.92}$ | $63.15_{\pm0.90}$ |
| **Hyperbolic-SGC** | $86.22_{\pm1.24}$ | $78.73_{\pm1.18}$ | $88.25_{\pm0.83}$ | $87.96_{\pm1.12}$ | $94.09_{\pm0.58}$ | $96.46_{\pm0.24}$ | $41.49_{\pm1.37}$ | $93.61_{\pm2.61}$ | $91.28_{\pm2.55}$ | $69.02_{\pm0.72}$ |
| *Improv.*($\Delta$) | ↑ 0.74 | ↓ 2.02 | ↑ 2.89 | ↓ 0.23 | ↑ 0.49 | ↑ 1.33 | ↑ 13.68 | ↑ 12.30 | ↑ 18.66 | ↑ 5.87 |
| APPNP | $88.14_{\pm0.73}$ | $80.47_{\pm0.74}$ | $88.12_{\pm0.31}$ | $85.32_{\pm0.37}$ | $88.51_{\pm0.31}$ | $95.83_{\pm0.32}$ | $39.66_{\pm0.55}$ | $90.98_{\pm1.64}$ | $91.81_{\pm1.96}$ | $68.53_{\pm0.36}$ |
| **Hyperbolic-APPNP** | $90.07_{\pm1.11}$ | $81.69_{\pm1.12}$ | $90.79_{\pm0.57}$ | $89.88_{\pm1.09}$ | $94.72_{\pm0.75}$ | $96.58_{\pm0.16}$ | $40.19_{\pm1.09}$ | $91.31_{\pm3.46}$ | $87.45_{\pm3.81}$ | $68.78_{\pm0.39}$ |
| *Improv.*($\Delta$) | ↑ 1.93 | ↑ 1.22 | ↑ 2.67 | ↑ 4.56 | ↑ 6.21 | ↑ 0.75 | ↑ 0.53 | ↑ 0.33 | ↓ 4.36 | ↑ 0.25 |
| GPR-GNN | $88.57_{\pm0.69}$ | $80.12_{\pm0.83}$ | $88.46_{\pm0.33}$ | $86.85_{\pm0.25}$ | $93.85_{\pm0.28}$ | $96.47_{\pm0.19}$ | $39.92_{\pm0.67}$ | $92.95_{\pm1.31}$ | $91.37_{\pm1.81}$ | $69.39_{\pm0.87}$ |
| **Hyperbolic-GPR** | $\mathbf{90.82_{\pm0.95}}$ | $\mathbf{81.88_{\pm1.33}}$ | $\mathbf{91.36_{\pm0.59}}$ | $88.21_{\pm0.50}$ | $94.49_{\pm0.63}$ | $96.58_{\pm0.19}$ | $40.68_{\pm0.85}$ | $93.28_{\pm3.58}$ | $92.77_{\pm3.36}$ | $68.62_{\pm0.62}$ |
| *Improv.*($\Delta$) | ↑ 2.25 | ↑ 1.76 | ↑ 2.90 | ↑ 1.36 | ↑ 0.64 | ↑ 0.11 | ↑ 0.76 | ↑ 0.33 | ↑ 1.40 | ↓ 0.77 |
| ChebNet | $86.67_{\pm0.82}$ | $79.11_{\pm0.75}$ | $87.95_{\pm0.28}$ | $87.54_{\pm0.43}$ | $93.77_{\pm0.32}$ | $95.66_{\pm0.28}$ | $37.61_{\pm0.89}$ | $86.22_{\pm2.45}$ | $83.93_{\pm2.13}$ | $68.88_{\pm0.81}$ |
| **Hyperbolic-Cheb** | $89.38_{\pm1.00}$ | $81.28_{\pm1.69}$ | $90.50_{\pm0.61}$ | $89.16_{\pm0.78}$ | $94.83_{\pm0.44}$ | $96.50_{\pm0.28}$ | $40.63_{\pm1.38}$ | $\mathbf{93.93_{\pm2.05}}$ | $91.06_{\pm4.46}$ | $68.75_{\pm0.63}$ |
| *Improv.*($\Delta$) | ↑ 2.71 | ↑ 2.17 | ↑ 2.65 | ↑ 1.62 | ↑ 1.06 | ↑ 0.84 | ↑ 3.02 | ↑ 7.71 | ↑ 7.13 | ↓ 0.13 |
| BernNet | $88.52_{\pm0.95}$ | $80.09_{\pm0.79}$ | $88.48_{\pm0.41}$ | $87.64_{\pm0.44}$ | $93.63_{\pm0.35}$ | $96.54_{\pm0.23}$ | $\mathbf{41.79_{\pm1.01}}$ | $93.12_{\pm0.65}$ | $92.13_{\pm1.64}$ | $68.67_{\pm0.55}$ |
| **Hyperbolic-Bern** | $88.34_{\pm0.92}$ | $80.27_{\pm0.76}$ | $89.52_{\pm0.80}$ | $88.61_{\pm0.93}$ | $94.24_{\pm0.87}$ | $96.64_{\pm0.28}$ | $39.96_{\pm0.98}$ | $93.11_{\pm1.29}$ | $89.79_{\pm3.45}$ | $68.74_{\pm0.75}$ |
| *Improv.*($\Delta$) | ↓ 0.18 | ↑ 0.18 | ↑ 1.04 | ↑ 0.97 | ↑ 0.61 | ↑ 0.10 | ↓ 1.83 | ↓ 0.01 | ↓ 2.34 | ↑ 0.07 |
| JacobiConv | $88.98_{\pm0.46}$ | $80.78_{\pm0.79}$ | $89.62_{\pm0.41}$ | $90.39_{\pm0.29}$ | $95.43_{\pm0.23}$ | $96.54_{\pm0.26}$ | $41.17_{\pm0.64}$ | $93.44_{\pm2.13}$ | $\mathbf{92.95_{\pm2.46}}$ | $68.66_{\pm0.92}$ |
| **Hyperbolic-Jacobi** | $89.69_{\pm1.83}$ | $81.17_{\pm1.63}$ | $90.17_{\pm0.66}$ | $88.69_{\pm0.79}$ | $94.53_{\pm0.75}$ | $96.58_{\pm0.25}$ | $39.73_{\pm0.84}$ | $91.97_{\pm4.19}$ | $90.64_{\pm5.13}$ | $68.76_{\pm0.96}$ |
| *Improv.*($\Delta$) | ↑ 0.71 | ↑ 0.39 | ↑ 0.55 | ↓ 1.70 | ↓ 0.90 | ↑ 0.04 | ↓ 1.44 | ↓ 1.47 | ↓ 2.31 | ↑ 0.10 |
| ChebNetII | $88.71_{\pm0.93}$ | $80.53_{\pm0.79}$ | $88.93_{\pm0.29}$ | $\mathbf{91.06_{\pm0.64}}$* | $\mathbf{95.88_{\pm0.58}}$* | $96.49_{\pm0.25}$ | $41.75_{\pm1.07}$ | $93.28_{\pm1.47}$ | $92.30_{\pm1.48}$ | $68.95_{\pm0.58}$ |
| **Hyperbolic-ChebII** | $90.39_{\pm0.95}$ | $81.60_{\pm1.38}$ | $90.34_{\pm0.77}$ | $89.82_{\pm0.68}$ | $94.53_{\pm0.46}$ | $\mathbf{96.67_{\pm0.23}}$ | $40.58_{\pm1.16}$ | $93.30_{\pm3.32}$ | $91.06_{\pm3.86}$ | $68.68_{\pm0.48}$ |
| *Improv.*($\Delta$) | ↑ 1.67 | ↑ 1.07 | ↑ 1.41 | ↓ 1.24 | ↓ 1.35 | ↑ 0.18 | ↓ 1.17 | ↑ 0.02 | ↓ 1.24 | ↓ 0.27 |

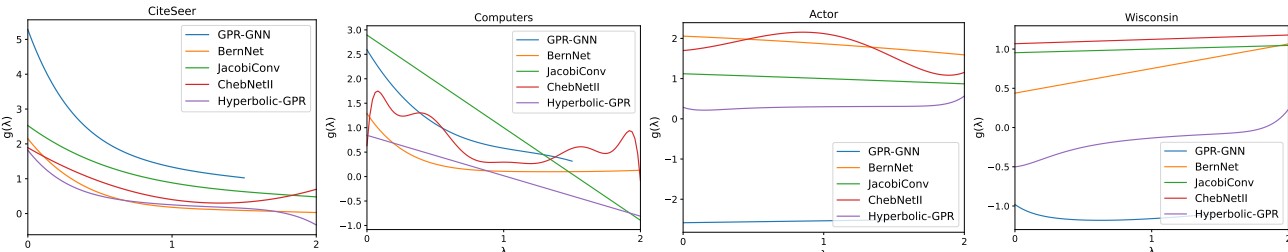

*Figure 1.* The graph spectral filter learned by the methods on graph datasets.

ditionally, our method also demonstrates competitive performance on PubMed and Actor. These results attest to the effectiveness of Hyperbolic-PDE GNN.

To further validate the performance improvement of modeling message passing as a system of hyperbolic partial differential equations, we choose several base models and modify them accordingly. Specifically, we let the polynomial $P(x)$ in Equation (15) to be the spectral graph convolution of the base model. Initially, we choose base models such as SGC, APPNP, and GPR-GNN based on monomials. Subsequently, we choose ChebNet, BernNet, JacobiConv, and ChebNetII as convolution forms, which are based on polynomials as basis. Table 5 indicate that methods based on monomials underperformed due to the insufficient ability of monomials to approximate filtering functions. In contrast, methods based on polynomials exhibit lower errors in function approximation, thus demonstrating superior performance. Upon enhancing all these methods to the system of hyperbolic PDEs, they all show varying degrees of performance improvement. Particularly for methods based on monomials, the introduction of the mechanism help to mitigate their

high error (further experiment can be found in Section 4.2. Our approach explicitly constrains the solution space of all aforementioned base models, ensuring that they are solely expressed by eigenvectors representing the topological structure and do not introduce additional features.

**Filter analysis.** Figure 1 visualizes the filters learned by some methods. We can observe that all methods exhibit powerful low-pass characteristics on CiteSeer and Computers. Particularly, due to the adoption of a Laplacian with self-loop in GPR-GNN, the absence of scaling coefficients in the higher frequency range results in a certain degree of weight explosion in the lower frequency range. Hyperbolic-GPR effectively mitigates this issue. On Actor and Wisconsin, the methods show weaker high-pass characteristics. Only the curve of Hyperbolic-GPR displays a trend of being high on the left and low on the right, indicating its amplification of high-frequency signals and suppression of low-frequency signals. Furthermore, due to the issues in GPR-GNN, when dealing with heterophilic graphs, it can only suppress low-frequency signals by assigning negative scaling coefficients

*Table 6.* Average of squared error ($R^2$ score) of methods over 50 images.

| | Low-pass | High-pass | Band-pass | Band-rejection | Comb | Low-band-pass | Runge |
|---|---|---|---|---|---|---|---|
| | $e^{-10\lambda^2}$ | $1-e^{-10\lambda^2}$ | $e^{-10(\lambda-1)^2}$ | $1-e^{-10(\lambda-1)^2}$ | $|\sin(\pi\lambda)|$ | $\begin{cases}1,\lambda<0.5\\e^{-100(\lambda-0.5)^2},0.5\leq\lambda<1\\e^{-50(\lambda-1.5)^2},\lambda\geq1\end{cases}$ | $1/(1+25\lambda^2)$ |
| GCN | 3.5364(.9871) | 67.1895(.2449) | 25.9193(.1061) | 20.9094(.9440) | 51.0312(.2933) | 15.5343(.9586) | 3.7937(.9860) |
| **Hyperbolic-GCN** | **1.3562(.9956)** | **1.7571(.9789)** | **4.4976(.7913)** | **3.0469(.9925)** | **8.8112(.8657)** | **4.6308(.9869)** | **0.6583(.9979)** |
| *Improv.*(%) | ↓**61.65%** | ↓**97.38%** | ↓**82.65%** | ↓**85.43%** | ↓**82.73%** | ↓**70.19%** | ↓**83.43%** |
| GAT | 2.6614(.9899) | 22.1274(.7457) | 13.7478(.4942) | 12.8713(.9650) | 22.1316(.6985) | 13.2587(.9634) | 2.8153(.9890) |
| **Hyperbolic-GAT** | **1.2671(.9963)** | **1.3592(.9822)** | **2.1492(.9152)** | **2.5862(.9933)** | **6.0365(.9055)** | **4.7609(.9866)** | **0.6192(.9979)** |
| *Improv.*(%) | ↓**52.39%** | ↓**93.86%** | ↓**84.37%** | ↓**79.71%** | ↓**72.72%** | ↓**64.09%** | ↓**78.01%** |
| ARMA | 1.8344(.9931) | 1.8304(.9793) | 7.6932(.7099) | 8.1734(.9785) | 15.2385(.7952) | 11.5748(.9677) | 2.1851(.9913) |
| **Hyperbolic-ARMA** | **0.9699(.9964)** | **1.0263(.9854)** | **1.7474(.9212)** | **1.6034(.9956)** | **5.9381(.9104)** | **4.9021(.9858)** | **0.4637(.9982)** |
| *Improv.*(%) | ↓**47.13%** | ↓**43.93%** | ↓**77.29%** | ↓**80.38%** | ↓**61.03%** | ↓**57.65%** | ↓**78.78%** |
| ChebNet | 0.8024(.9973) | 0.7877(.9903) | 2.3232(.9090) | 2.5446(.9934) | 4.0797(.9443) | 4.4431(.9878) | 0.4048(.9988) |
| **Hyperbolic-Cheb** | **0.0077(1.000)** | **0.0088(.9999)** | **0.0567(.9978)** | **0.1077(.9997)** | **0.2019(.9969)** | **0.2276(.9994)** | **0.0169(.9999)** |
| *Improv.*(%) | ↓**99.04%** | ↓**98.88%** | ↓**97.56%** | ↓**95.77%** | ↓**97.50%** | ↓**94.88%** | ↓**95.83%** |
| BernNet | 0.0309(.9999) | **0.0103(.9998)** | 0.0477(.9982) | 0.9434(.9973) | 1.1073(.9854) | 2.9633(.9917) | 0.0346(.9999) |
| **Hyperbolic-Bern** | **0.0045(1.000)** | 0.0138(.9998) | **0.0154(.9992)** | **0.1532(.9996)** | **0.1921(.9973)** | **0.6738(.9982)** | **0.0256(.9999)** |
| *Improv.*(%) | ↓**85.44%** | ↑**33.98%** | ↓**67.71%** | ↓**83.76%** | ↓**82.65%** | ↓**77.26%** | ↓**26.01%** |
| ChebNetII | 0.0068(1.000) | 0.0028(1.000) | 0.0980(.9760) | 0.0162(1.000) | **0.2832(.9964)** | 1.3945(.9964) | 0.0089(.9999) |
| **Hyperbolic-ChebII** | **0.0039(1.000)** | **0.0017(.9999)** | **0.0099(.9991)** | **0.0125(1.000)** | 0.2918(.9962) | 1.3800(.9962) | **0.0059(1.000)** |
| *Improv.*(%) | ↓**42.65%** | ↓**39.29%** | ↓**89.90%** | ↓**22.22%** | ↑**3.04%** | ↓**1.04%** | ↓**33.71%** |

| a) Low-pass | b) High-pass | c) Band-pass | d) Band-rejection | e) Comb | f) Low-band-pass | g) Runge |
|---|---|---|---|---|---|---|

*Figure 2.* The image signals processed by the real filter (first row), Hyperbolic-GCN (second row), and GCN (third row).

to them. Hyperbolic-GPR also circumvents this problem.

## 4.2. Signal Filtering on Images

In this section, we investigate the fitting capability of polynomial spectral filters under the influence of Hyperbolic-PDE GNN. We conduct experiment on 50 real images with a resolution of $100 \times 100$ from the Image Processing Toolbox in Matlab. Here, we transform the images into a 2D regular 4-neighborhood grid graph following the experimental setup

in (Balcilar et al., 2021; He et al., 2021), where each pixel denotes a node. Each image is associated with a $100 \times 100$ adjacency matrix and a 10000-dimensional signal vector **x** that records pixel intensity.

For each image, we filter its original signals using 7 following spectral filters: low-pass filter, high-pass filter, band-pass filter, band-rejection filter, comb filter, low-band-pass filter, and runge filter. The formula and its curve of each filter can be found in Table 6. For example, applying low-

pass filter to an image is equivalent to performing operation $\mathbf{y} = \mathbf{U}\mathrm{diag}\{e^{-10\lambda_1^2}, \ldots, e^{-10\lambda_N^2}\}\mathbf{U}^\top \mathbf{x}$ on graph signal, where the filtered graph signal $\mathbf{y}$ is used as the ground truth.

The objective of this experiment is to minimize the average of of squared error between the model output and ground truth across 50 images. Therefore, this experiment is considered as a regression task. We utilize the coefficient of determination (*i.e.*, $R^2$ score, closer to 1 is better) to reflect the degree of fit in regression. We select GCN (Kipf & Welling, 2017), GAT (Velickovic et al., 2018), ARMA (Bianchi et al., 2021), GPR-GNN (Chien et al., 2021), ChebNet (Defferrard et al., 2016), BernNet (He et al., 2021), JacobiConv (Wang & Zhang, 2022), and ChebNetII (He et al., 2022) as baselines and evaluate their fitting capabilities under the enhancement of Hyperbolic-PDE GNN. To ensure fairness, we follow the setup of He et al. (2021) to reproduce the results of each baseline. For our method, we conduct parameter search using wandb. *See appendix C for specific experimental configurations.*

As shown in Table 6, GCN and GAT exhibit good properties as low-pass filters, but struggle to fit more complex filters. On the other hand, polynomial spectral filters have the ability to fit complex filters. All the aforementioned base models demonstrate enhanced fitting capabilities under the augmentation of Hyperbolic-PDE GNN. Particularly notable is the significant improvement observed in GCN and GAT, indicating that Hyperbolic-PDE GNN not only enhances the expressive power of spectral filters but also endows them with the ability to fit complex filters.

**Filter analysis.** Figure 2 illustrates graph signals obtained through real filters, Hyperbolic-GCN, and GCN. We can observe that the graph signals filtered by Hyperbolic-GCN closely resemble the original graph signals, while GCN struggles to extract useful features when fitting complex filters. Moreover, the most notable improvement is seen in ChebNet, whose inferior performance is attributed to learning illegal filter coefficients (He et al., 2022). Hyperbolic-Cheb can limit the solution space by defining initial values, thereby reducing the risk of learning illegal coefficients.

## 5. Related Works

**Spectral Graph Neural Networks.** Spectral CNN (Bruna et al., 2014), as a pioneering work in spectral graph neural networks, first apply spectral graph theory to the field of graphs. GCN (Kipf & Welling, 2017), as the most classic subsequent work, behaves like a low-pass filter. SGC (Wu et al., 2019) builds upon GCN by removing non-linear activation functions and simplifying the parameter matrices in each layer, demonstrating that linear GNNs also exhibit strong performance. APPNP (Klicpera et al., 2019a) proposes an improved propagation scheme based on Person-

alized PageRank. These early works focus on exploring features of homophilic graphs but show poor performance on heterophilic graphs. GPR-GNN (Chien et al., 2021) designs filter parameters based on PageRank to adaptively learn low-pass or high-pass filters. GNN-LF/HF (Zhu et al., 2021) uses adjustable graph kernels to design low-pass and high-pass filters. ARMA (Bianchi et al., 2021) employs auto-regressive moving average filters to design graph convolutions, which can better capture global graph features. While these methods can adapt to different types of graphs, their performance suffers when fitting complex filters due to their design of graph filters based on monomials. ChebNet (Defferrard et al., 2016), as an early attempt of polynomial filters, first approximates graph filters using Chebyshev polynomials as a basis. BernNet (He et al., 2021) utilizes Bernstein polynomials as a basis to simulate complex filters such as band-pass and comb filters. JacobiConv (Wang & Zhang, 2022) unifies some methods based on orthogonal polynomials, like ChebNet, by employing more general Jacobi polynomials. ChebNetII (He et al., 2022) addresses the issue of learning Illegal filter parameters in ChebNet by introducing Chebyshev interpolation to reduce approximation errors in Chebyshev polynomials. Most of these methods construct graph filters based on mature polynomial basis in mathematics, theoretically capable of approximating any filter function. Another class of methods designs more targeted filters to address various graph problems. G2CN (Li et al., 2022) introduces Gaussian graph filters to enable models to flexibly adapt to different types of graphs. Even-Net (Lei et al., 2022) proposes even-polynomial graph filters to enhance the model's robustness on heterophilic graphs. OptBasisGNN (Guo & Wei, 2023) learn the optimal basis in the orthogonal polynomial space. UniFilter (Huang et al., 2024b) specifically designs adaptive heterophily basis for heterophilic graphs. NFGNN (Zheng et al., 2024) addresses the challenge of local patterns in graphs by proposing local spectral filters. AdaptKry (Huang et al., 2024a) optimizes graph filters using basis from adaptively Krylov subspaces. All these methods are defined in the original spatial domain, meaning the node features they obtain lack an interpretable solution space to explain their relationship with topological features, thus lacking necessary interpretability.

**Differential Equation on Graph.** In recent years, research has gradually delved into the dynamical systems of graph neural networks. Neural ODE (Chen et al., 2018) initially models embeddings as a continuous dynamic model based on ordinary differential equations (ODE). GDE (Poli et al., 2019) employs forward Euler method to solve ODE on graphs. CGNN (Xhonneux et al., 2020) directly computes the analytical solution of ODE. GraphCON (Rusch et al., 2022) establishes a connection with conventional graph convolutions based on second-order ODE. Kuramo-toGNN (Nguyen et al., 2024) specifically utilizes the Ku-

ramoto model to address over-smoothing issues. MGKN (Li et al., 2020) introduces partial differential equations (PDE) into graph problems. PDE-GCN (Eliasof et al., 2021) constructs graph convolutions using diffusion and hyperbolic equations, respectively. DGC (Wang et al., 2021) decouples terminal time from propagation steps. GDC (Klicpera et al., 2019b) proposes graph diffusion convolution to extract information from indirectly connected neighbors. ADC (Zhao et al., 2021) further introduces adaptively diffusion convolution to aggregate neighbors within the optimal neighborhood. Subsequent models like GRAND (Chamberlain et al., 2021), GRAND++ (Thorpe et al., 2022), and HiD-Net (Li et al., 2024) treat message passing as a heat diffusion process and model it based on the heat diffusion equation. Most of the aforementioned methods heuristically integrate differential equations into graph neural networks and demonstrate empirically their effectiveness. However, these methods do not systematically explore why models based on differential equations perform well, and do not delve into the intrinsic properties of node features obtained these methods.

**Discussion with Hyperbolic Geometry GNNs** The hyperbolic PDEs and hyperbolic geometry (Liu et al., 2019; Chami et al., 2019) exhibit fundamentally distinct characteristics. Specifically, *hyperbolic geometry* focuses on the hierarchical structure of graphs, which typically involves mapping nodes to hyperbolic space using models such as Poincaré Ball models or Lorentz models to extract hierarchical relationships between nodes. On the other hand, our focus of *hyperbolic PDEs* lies in constructing the message passing mechanism of GNNs based on the theory of PDE. This design naturally maps nodes into a solution space spanned by a set of eigenvectors of the graph, which can improve the ability to learn complex graph filters.

## 6. Conclusion

To resolve limitation of the spatial space of MP in representing node semantic features, this paper introduces a Hyperbolic-PDE GNN based on a system of hyperbolic PDEs. We theoretically demonstrate that the solution space is spanned by orthogonal bases that describe the topological structure of graphs, enabling to represent the topological features, thereby significantly increasing the interpretability of the model. Moreover, we enhance the flexibility and robustness of the model using polynomial spectral filters and establish a connection between hyperbolic PDEs and spectral GNNs. Extensive experiments show that our approach exhibits effective and robust results with traditional GNNs.

## Acknowledgments

This work is supported by the National Natural Science Foundation of China (No.62406319), the Youth Innovation Promotion Association of CAS (Grant No. 2021153), and the Postdoctoral Fellowship Program of CPSF (No.GZC20232968).

## Impact Statement

The work presented in this paper aims to advance the field of graph deep learning. We do not expect any immediate, negative societal impact of the present work. It delves into the theory of graph neural networks, and thus may benefit the applications that involve graphs by making them more effective and reliable.

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

# A. Theoretical Analysis

## A.1. Detailed Derivation of the System of Partial Differential Equations

Given Equation (8), we combine the equations of all nodes on any dimension $l$ to obtain a system of second-order linear partial differential equations:

$$\begin{cases} \dfrac{\partial^2 x_{1l}}{\partial t^2} = a^2 \displaystyle\sum_{v_j \in \mathcal{N}(v_1)} (x_{1l} - x_{jl}), \\[2ex] \dfrac{\partial^2 x_{2l}}{\partial t^2} = a^2 \displaystyle\sum_{v_j \in \mathcal{N}(v_2)} (x_{2l} - x_{jl}), \\[2ex] \cdots \\[1ex] \dfrac{\partial^2 x_{nl}}{\partial t^2} = a^2 \displaystyle\sum_{v_j \in \mathcal{N}(v_n)} (x_{nl} - x_{jl}). \end{cases} \tag{21}$$

This system of equations represents the evolution pattern of a graph signal on any dimension $l$ over time. Next, we extend the dimension to $d$ dimensions:

$$\begin{cases} \dfrac{\partial^2 \mathbf{x}_1}{\partial t^2} = [\dfrac{\partial^2 x_{11}}{\partial t^2}, \dfrac{\partial^2 x_{12}}{\partial t^2}, \ldots, \dfrac{\partial^2 x_{1d}}{\partial t^2}], \\[2ex] \dfrac{\partial^2 \mathbf{x}_2}{\partial t^2} = [\dfrac{\partial^2 x_{21}}{\partial t^2}, \dfrac{\partial^2 x_{22}}{\partial t^2}, \ldots, \dfrac{\partial^2 x_{2d}}{\partial t^2}], \\[2ex] \cdots \\[1ex] \dfrac{\partial^2 \mathbf{x}_n}{\partial t^2} = [\dfrac{\partial^2 x_{n1}}{\partial t^2}, \dfrac{\partial^2 x_{n2}}{\partial t^2}, \ldots, \dfrac{\partial^2 x_{nd}}{\partial t^2}]. \end{cases} \tag{22}$$

Then, the right-hand side of Equation (21) can be extended as

$$a^2 \begin{bmatrix} \sum_{v_j \in \mathcal{N}(v_1)} (x_{11} - x_{j1}) & \sum_{v_j \in \mathcal{N}(v_1)} (x_{12} - x_{j2}) & \cdots & \sum_{v_j \in \mathcal{N}(v_1)} (x_{1d} - x_{jd}) \\ \sum_{v_j \in \mathcal{N}(v_2)} (x_{21} - x_{j1}) & \sum_{v_j \in \mathcal{N}(v_2)} (x_{22} - x_{j2}) & \cdots & \sum_{v_j \in \mathcal{N}(v_2)} (x_{2d} - x_{jd}) \\ \vdots & \vdots & \ddots & \vdots \\ \sum_{v_j \in \mathcal{N}(v_n)} (x_{n1} - x_{j1}) & \sum_{v_j \in \mathcal{N}(v_n)} (x_{n2} - x_{j2}) & \cdots & \sum_{v_j \in \mathcal{N}(v_n)} (x_{nd} - x_{jd}) \end{bmatrix}. \tag{23}$$

By combining Equation (22) and Equation (23), we obtain:

$$\frac{\partial^2 \mathbf{X}}{\partial t^2} = a^2 [\sum_{v_j \in \mathcal{N}(v_1)} (\mathbf{x}_1 - \mathbf{x}_j), \sum_{v_j \in \mathcal{N}(v_2)} (\mathbf{x}_2 - \mathbf{x}_j), \ldots, \sum_{v_j \in \mathcal{N}(v_n)} (\mathbf{x}_n - \mathbf{x}_j)] = a^2 \widehat{\mathbf{L}} \mathbf{X}, \tag{24}$$

where $\mathbf{x}_i = [x_{i1}, x_{i2}, \ldots, x_{il}]^\top$.

## A.2. Proof of Theorem 3.1

Given Equation (9), we consider a graph signal on any dimension $l$ (*i.e.*, Equation (21)).

Let $\frac{\partial x_{1l}}{\partial t} = y_{1l}, \frac{\partial x_{2l}}{\partial t} = y_{2l}, \ldots, \frac{\partial x_{nl}}{\partial t} = y_{nl}$, then $\frac{\partial y_{1l}}{\partial t} = a^2 \sum_{v_j \in \mathcal{N}(v_1)} (x_{1l} - x_{jl}), \frac{\partial y_{2l}}{\partial t} = a^2 \sum_{v_j \in \mathcal{N}(v_2)} (x_{2l} - x_{jl}), \ldots, \frac{\partial y_{nl}}{\partial t} = a^2 \sum_{v_j \in \mathcal{N}(v_n)} (x_{nl} - x_{jl})$.

Subsequently, let $\mathbf{w}_l = [\mathbf{y}_{:l}, \mathbf{x}_{:l}] \in \mathbb{R}^{2n}$, Equation (21) can be rewritten in the following form:

$$\frac{d\mathbf{w}_l}{dt} = \mathbf{C}\mathbf{w}_l = \begin{bmatrix} \mathbf{I}_n & \mathbf{0} \\ \mathbf{0} & a^2\widehat{\mathbf{L}} \end{bmatrix} \mathbf{w}_l, \tag{25}$$

where $\widehat{\mathbf{L}} = \mathbf{D} - \mathbf{A} \in \mathbb{R}^{n \times n}$ is a the Laplacian matrix.

Note that since spatial derivatives w.r.t space have been transformed into gradients and divergences of graph, $\mathbf{w}_l$ is now only differentiated w.r.t time. Therefore, Equation (25) essentially is a system of first-order constant-coefficient homogeneous linear differential equations.

To obtain the fundamental matrix of solution of Equation (25), it is necessary to introduce the following definition and theorem from the mathematical domain.

Given a system of first-order homogeneous linear differential equations

$$\frac{d\mathbf{x}}{dt} = \mathbf{C}(t)\mathbf{x}, \tag{26}$$

we have:

**Definition A.1.** If each column of a matrix $\mathbf{\Phi}(t) \in \mathbb{R}^{n \times n}$ is a solution to Equation (26), then the matrix $\mathbf{\Phi}(t)$ is referred to as the **solution matrix** of Equation (26). If the columns of the solution matrix are linearly independent over the interval $a \leq t \leq b$, it is referred to as the **fundamental matrix of solution** of Equation (26) on the interval $a \leq t \leq b$.

Here, we represent $n$ linearly independent column vectors from the matrix $\mathbf{\Phi}(t)$ as $\boldsymbol{\varphi}_1(t), \boldsymbol{\varphi}_2(t), \ldots, \boldsymbol{\varphi}_n(t)$. According to Definition A.1, we can deduce the following theorems:

**Theorem A.2.** *The fundamental matrix of solution defines the **solution space** of Equation (26), enabling us to express any solution of Equation (26) as a linear combination of $n$ linearly independent column vectors $\boldsymbol{\varphi}_1(t), \boldsymbol{\varphi}_2(t), \ldots, \boldsymbol{\varphi}_n(t)$ from the fundamental matrix of solution $\mathbf{\Phi}(t)$.*

Now consider a system of first-order constant-coefficient homogeneous linear differential equations

$$\frac{d\mathbf{x}}{dt} = \mathbf{C}\mathbf{x}, \tag{27}$$

where $\mathbf{C} \in \mathbb{R}^{n \times n}$ is a constant matrix.

**Theorem A.3.** *The matrix*

$$\mathbf{\Phi}(t) = \exp \mathbf{C}t \tag{28}$$

*is a fundamental matrix of solution of Equation (27), and $\mathbf{\Phi}(0) = \mathbf{I}$.*

*Proof.* It is easily inferred from the definition that $\mathbf{\Phi}(0) = \mathbf{I}$. Taking the derivative of Equation (28) w.r.t $t$, we obtain:

$$\begin{aligned}
\frac{d\mathbf{\Phi}(t)}{dt} &= \frac{d}{dt} \exp \mathbf{C}t \\
&= \mathbf{C} + \frac{\mathbf{C}^2 t}{1!} + \frac{\mathbf{C}^3 t^2}{2!} + \cdots + \frac{\mathbf{A}^k t^{k-1}}{(k-1)!} + \cdots \\
&= \mathbf{C} \exp \mathbf{C}t \\
&= \mathbf{C}\mathbf{\Phi}(t)
\end{aligned} \tag{29}$$

This indicates that $\mathbf{\Phi}(t)$ is the solution matrix of Equation (27). Furthermore, because $\det \mathbf{\Phi}(0) = \det \mathbf{I} = 1$, $\mathbf{\Phi}(t)$ is the fundamental matrix of solution of Equation (27). $\qquad \square$

Therefore, for Equation (25), its fundamental matrix of solution is

$$\begin{aligned}
\mathbf{\Phi}(t) &= \exp \mathbf{C}t \\
&= \mathbf{I} + \begin{bmatrix} \mathbf{I} & \mathbf{0} \\ \mathbf{0} & a^2\widehat{\mathbf{L}} \end{bmatrix} \frac{t}{1!} + \begin{bmatrix} \mathbf{I} & \mathbf{0} \\ \mathbf{0} & a^2\widehat{\mathbf{L}}^2 \end{bmatrix} \frac{t^2}{2!} + \cdots + \begin{bmatrix} \mathbf{I} & \mathbf{0} \\ \mathbf{0} & a^2\widehat{\mathbf{L}}^k \end{bmatrix} \frac{t^k}{k!} + \cdots \\
&= \begin{bmatrix} \exp \mathbf{I}t & \mathbf{0} \\ \mathbf{0} & a^2 \exp \widehat{\mathbf{L}}t \end{bmatrix}.
\end{aligned} \tag{30}$$

where $\boldsymbol{\varphi}_1(t), \boldsymbol{\varphi}_2(t), \ldots, \boldsymbol{\varphi}_{2n}(t)$ are the column vectors from $\mathbf{\Phi}(t)$.

Ultimately, the solution space of all dimension can be represented as the Cartesian product of the solution spaces of each dimension:

$$\mathbf{\Psi} = \underbrace{\mathbf{\Phi} \times \mathbf{\Phi} \times \cdots \times \mathbf{\Phi}}_{d}\Phi, \tag{31}$$

where $\times$ is the operator of Cartesian product.

## A.3. Proof of Theorem 3.3

*Proof.* Let $\boldsymbol{\varphi}(t) = e^{\lambda t}\mathbf{u}, \mathbf{u} \neq \mathbf{0}$, and substituting it into Equation (25), we obtain:

$$\lambda e^{\lambda t}\mathbf{u} = \mathbf{C}e^{\lambda t}\mathbf{u}. \tag{32}$$

Because $e^{\lambda t} \neq 0$, Equation (32) becomes:

$$(\lambda\mathbf{I} - \mathbf{C})\mathbf{u} = \mathbf{0}. \tag{33}$$

We can observe that Equation (33) is the characteristic equation of matrix $\mathbf{C}$. Therefore, $e^{\lambda t}\mathbf{u}$ constitutes a solution to Equation (25) if and only if $\lambda$ is the eigenvalue of $\mathbf{C}$ and $\mathbf{u}$ is the corresponding eigenvector.

For the matrix $\mathbf{C}$ in Equation (25), its characteristic equation is

$$\det(\lambda\mathbf{I} - \mathbf{C}) = |\lambda\mathbf{I} - \mathbf{C}| = \begin{vmatrix} \lambda\mathbf{I} - \mathbf{I} & \mathbf{0} \\ \mathbf{0} & \lambda\mathbf{I} - a^2\widehat{\mathbf{L}} \end{vmatrix} = |\lambda\mathbf{I} - \mathbf{I}||\lambda\mathbf{I} - a^2\widehat{\mathbf{L}}| = 0. \tag{34}$$

Therefore, the eigenvalues of $\mathbf{I}$ and $a^2\widehat{\mathbf{L}}$ are both eigenvalues of $\mathbf{C}$, that is $\lambda_1' = \lambda_2' = \cdots = \lambda_n' = 1, \lambda_{n+1}' = a^2\widehat{\lambda}_1, \lambda_{n+2}' = a^2\widehat{\lambda}_2, \ldots, \lambda_{2n}' = a^2\widehat{\lambda}_n$.

The corresponding eigenvectors of $\mathbf{I}$ are $\mathbf{v}_1 = [1, 0, \ldots, 0]^\top, \mathbf{v}_2 = [0, 1, \ldots, 0]^\top, \ldots, \mathbf{v}_n = [0, 0, \ldots, 1]^\top$, and the corresponding eigenvectors of $\widehat{\mathbf{L}}$ are $\widehat{\mathbf{u}}_1, \widehat{\mathbf{u}}_2, \ldots, \widehat{\mathbf{u}}_n$, thus the eigenvectors of $\mathbf{C}$ are

$$\mathbf{u}_1' = \begin{bmatrix} \mathbf{v}_1 \\ \mathbf{0} \end{bmatrix}, \mathbf{u}_2' = \begin{bmatrix} \mathbf{v}_2 \\ \mathbf{0} \end{bmatrix}, \ldots, \mathbf{u}_n' = \begin{bmatrix} \mathbf{v}_n \\ \mathbf{0} \end{bmatrix}, \mathbf{u}_{n+1}' = \begin{bmatrix} \mathbf{0} \\ \widehat{\mathbf{u}}_1 \end{bmatrix}, \mathbf{u}_{n+2}' = \begin{bmatrix} \mathbf{0} \\ \widehat{\mathbf{u}}_2 \end{bmatrix}, \ldots, \mathbf{u}_{2n}' = \begin{bmatrix} \mathbf{0} \\ \widehat{\mathbf{u}}_n \end{bmatrix}. \tag{35}$$

Because $\mathbf{u}_1', \mathbf{u}_2', \ldots, \mathbf{u}_{2n}'$ are linearly independently, matrix

$$\boldsymbol{\Phi}(t) = [e^{\lambda_1 t}\mathbf{u}_1', e^{\lambda_2 t}\mathbf{u}_2', \ldots, e^{\lambda_{2n} t}\mathbf{u}_{2n}'], \tag{36}$$

is a fundamental matrix of solution of Equation (25).

Because both $\exp\mathbf{C}t$ and $\boldsymbol{\Phi}(t)$ are solutions of Equation (25), there exists a non-singular constant matrix $\mathbf{B}$ such that

$$\exp\mathbf{C}t = \boldsymbol{\Phi}(t)\mathbf{B}. \tag{37}$$

Let $t = 0$, we have $\mathbf{B} = \boldsymbol{\Phi}^{-1}(0)$. Therefore,

$$\exp\mathbf{C}t = \boldsymbol{\Phi}(t)\boldsymbol{\Phi}^{-1}(0). \tag{38}$$

$\square$

# B. Details of Polynomial-based Spectral Graph Neural Networks

**SGC.** Wu et al. (2019) directly utilizes monomials as filter

$$g(\widetilde{\lambda}) = (1 - \widetilde{\lambda})^K, \ \widetilde{\lambda} \in [0, 2) \tag{39}$$

to obtain the spectral graph convolution

$$\mathbf{Z} = (\mathbf{I} - \widetilde{\mathbf{L}})^K \mathbf{X}\mathbf{W}. \tag{40}$$

**APPNP.** Klicpera et al. (2019a) leverage the concept of personalized PageRank (Page et al., 1999) to design graph convolutions, where its filter is equivalent to a polynomial

$$g(\widetilde{\lambda}) = \sum_{k=0}^{K} \frac{\alpha^k}{1 - \alpha}(1 - \widetilde{\lambda})^k, \ \widetilde{\lambda} \in [0, 2) \tag{41}$$

with monomial basis $(1 - \widetilde{\lambda})^k$. Then the spectral graph convolution of APPNP is given by:

$$\mathbf{Z} = \alpha(\mathbf{I} - (1 - \alpha)(\mathbf{I} - \widetilde{\mathbf{L}}))^{-1}\phi(\mathbf{X}), \tag{42}$$

where $\alpha$ is a hyperparameter, so the filter cannot be learned during the training process.

**GPR-GNN.** Chien et al. (2021) employ learnable parameters $\alpha_k$ as coefficients to learn a polynomial filter

$$g(\widetilde{\lambda}) = \sum_{k=0}^{K} \theta_k (1 - \widetilde{\lambda})^k, \ \widetilde{\lambda} \in [0, 2) \tag{43}$$

with monomial basis like APPNP, can be viewed as a Generalized PageRank. Then the spectral graph convolution of GPR-GNN is given by:

$$\mathbf{Z} = \sum_{k=0}^{K} \theta_k (\mathbf{I} - \widetilde{\mathbf{L}})^k \phi(\mathbf{X}) \tag{44}$$

**ChebNet.** Defferrard et al. (2016) approximate graph spectral filter using Chebyshev polynomial approximation

$$g(\lambda) = \sum_{k=0}^{K-1} \theta_k T_k(\lambda - 1), \ \lambda \in [0, 2]. \tag{45}$$

Here, for $k = 0, \ldots, K$, $T_0(\lambda) = 1, T_1(\lambda) = \lambda$, $T_k(\lambda) = 2\lambda T_{k-1}(\lambda) - T_{k-2}(\lambda)$ are the Chebyshev base. Then the spectral graph convolution of ChebNet is given by:

$$\mathbf{Z} = \sum_{k=0}^{K-1} T_k(\mathbf{L} - \mathbf{I})\mathbf{X}\mathbf{W}_k. \tag{46}$$

**BernNet.** He et al. (2021) utilize Bernstein polynomial approximation to approximate graph spectral filters:

$$g(\lambda) = \sum_{k=0}^{K} \theta_k b_{k,K}(\lambda/2), \ \lambda \in [0, 2]. \tag{47}$$

Here, for $k = 0, \ldots, K$, $b_{k,K}(\lambda) = \binom{K}{k}(1 - \lambda)^{(K-k)}\lambda^k, \lambda \in [0, 1]$ are the Bernstein base. The Bernstein polynomial approximation can learn arbitrary spectral filters. Then the spectral graph convolution of BernNet is given by:

$$\mathbf{Z} = \sum_{k=0}^{K} \frac{\theta_k}{2^K} \binom{K}{k} (2\mathbf{I} - \mathbf{L})^{K-k} \mathbf{L}^k \phi(\mathbf{X}). \tag{48}$$

**JacobiConv.** Wang & Zhang (2022) approximate spectral graph convolutions using Jacobi polynomial approximation:

$$g(\lambda) = \sum_{k=0}^{K} \theta_k P_k^{a,b}(1 - \lambda), \ \lambda \in [0, 2]. \tag{49}$$

Here, for $k = 0, \ldots, K$, $P_0^{(a,b)}(\lambda) = 1, P_1^{(a,b)}(\lambda) = \frac{a-b}{2} + \frac{a+b+2}{2}\lambda, P_k^{(a,b)}(\lambda) = (\alpha_k\lambda + \alpha_k')P_{k-1}^{(a,b)}(\lambda) - \alpha_k''P_{k-2}^{(a,b)}(\lambda)$ are the Jacobi base, where $a, b$ are hyperparameters and $\beta_k = \frac{(2k+a+b)(2k+a+b-1)}{2k(k+a+b)}, \alpha_k' = \frac{(2k+a+b-1)(a^2-b^2)}{2k(k+a+b)(2k+a+b-2)}, \alpha_k'' = \frac{(k+a-1)(k+b-1)(2k+a+b)}{k(k+a+b)(2k+a+b-2)}$. The Jacobi polynomial is a general form of Chebyshev polynomials (*i.e.* $a = b = 0.5$). Then the spectral graph convolution of JaocibConv is given by:

$$\mathbf{Z}_{:j} = \sum_{i=0}^{K} \theta_{kj} P_k^{a,b}(\mathbf{I} - \mathbf{L})\phi(\mathbf{X}_{:j}). \tag{50}$$

**ChebNetII.** He et al. (2022) discover that the poor performance of Chebyshev polynomials is due to the learned illegal parameters triggering the Runge phenomenon (where the fitted function deviates from the original function as the order increases). Subsequently, they introduce Chebyshev interpolation into Chebyshev polynomials to enhance graph spectral filters:

$$g(\lambda) = \sum_{k=0}^{K} c_k(\theta) T_k(\lambda - 1), \ \lambda \in [0, 2]. \tag{51}$$

Here, the coefficients $c_0(\theta) = \frac{1}{K+1} \sum_{j=0}^{K} \alpha_j, c_k(\theta) = \frac{2}{K+1} \sum_{j=0}^{K} \theta_j T_k(x_j)$ are obtained from Chebyshev nodes $x_j = \cos((j+1/2)\pi/(K+1))$. Then the spectral graph convolution of ChebNetII is given by:

$$\mathbf{Z} = \frac{2}{K+1} \sum_{k=0}^{K} \sum_{j=0}^{K} \theta_j T_k(x_j) T_k(\mathbf{L} - \mathbf{I})\phi(\mathbf{X}). \tag{52}$$

## C. Detailed Experimental Setup

**Parameter search.** We utilize wandb library for parameter search in node classification task. The search strategy adopt the Bayes strategy, with parameter ranges as shown in Table 7.

*Table 7.* The range of hyperparameters in node classification task.

| Hyperparameters | Range | Distribution |
|---|---|---|
| Hidden-dimension $d$ | $\{32, 64, 128\}$ | values |
| Polynomial order $K$ | $\{1, 2, 3, 4, 5, 6, 7, 8, 9, 10\}$ | values |
| Time size $\tau$ | $\{0.2, 0.5, 1.0, 2.0, 5.0\}$ | values |
| Terminal time $T$ | $[1, 20]$ | uniform |
| Dropout | $[0, 0.8]$ | uniform |
| Learning rate | $[0.001, 0.25]$ | log_uniform_values |
| Weight decay | $[0, 0.1]$ | uniform |

Furthermore, we conduct parameter search in signal filtering task according to the schemes outlined in Table 8. For Hyperbolic-GCN, Hyperbolic-GAT, and Hyperbolic-ARMA, we only utilize scheme 1. For Hyperbolic-Cheb, Hyperbolic-Bern, and Hyperbolic-ChebII, to minimize the number of parameter combinations, we respectively employ scheme 1 and scheme 2 for search and select the optimal parameter combination from them.

*Table 8.* The range of hyperparameters in signal filtering task.

| Hyperparameters | Scheme 1 | | Scheme 2 | |
|---|---|---|---|---|
| | Range | Distribution | Range | Distribution |
| Polynomial order $K$ | $\{1, 2, \ldots, 10\}$ | values | 10 | value |
| Time size $\tau$ | $\{0.5, 0.75, \ldots, 5\}$ | values | $\{0.5, 1\}$ | values |
| Terminal time $T$ | $[1, 10]$ | uniform | $\{0.5, 1, 2\}$ | values |
| Learning rate | $[0.001, 0.25]$ | log_uniform_values | $[0.001, 0.25]$ | log_uniform_values |

**Training process.** In the node classification task, we employ cross-entropy as the loss function and utilize the Adam optimizer (Kingma & Ba, 2015) for optimization. The models are trained for 200 epochs with early stopping set to 10. In the signal filtering task, we optimize the squared error between the model output and ground truth using the Adam optimizer. The models are trained for 2000 epochs with early stopping set to 100.

