# OpenReview forum: "Hyperbolic-PDE GNN: Spectral Graph Neural Networks in the Perspective of A System of Hyperbolic Partial Differential Equations"
_ICML.cc/2025/Conference — ICML 2025 poster_

### Official Review · Reviewer_4yEq · 2025-02-27

**Overall Recommendation:** 4

**Summary:**

The paper introduces Hyperbolic GNN, a novel method that models a graph as a system of differential equations by leveraging the good properties of hyperbolic differential equations. It incorporates the topological structure characteristics of the graph into the message passing by modeling the differential equations, and provides a complete theoretical proof for the solution space of the constructed system. To address the computational complexity caused by Laplacian, polynomial theory is introduced to improve the original Laplacian and enhance the nonlinearity of the filter. Compared to state-of-the-art methods, the approach exhibits excellent performance on a large number of graph datasets and image datasets. This paper demonstrates a well-organized logical structure and rigorous theoretical proofs, making a significant contribution to the theoretical research on GNN within the paradigm of differential equations.

**Claims And Evidence:**

This paper presents the whole process of the proposed Hyperbolic GNN through a specific hyperbolic equation. The graph is modeled as a system of differential equations, and the coefficient matrix is completely determined by the topological structure of the graph itself. Based on the ordinary differential equation theory, the existence and spatial structure of the solution of the system are completely proved, and the original Laplacian of the system is approximated by the polynomial theory.

**Essential References Not Discussed:**

The work highly related to this paper is mainly spectral GNNs based on the differential equation (ODEs, PDEs) paradigm. The necessary articles have been cited in this paper.

**Experimental Designs Or Analyses:**

The experiments are mostly convincible, which mainly consist of three parts:
1. The performance of Hyperbolic GNN is compared with common spectral graph networks on graph datasets.
2. Based on hyperbolic partial differential equations, it is analyzed that the base enhances the performance of traditional spectral GNNs, and the effectiveness of polynomial approximation is verified.
3. Interestingly, the designed Hyperbolic GNN is applied to image filtering experiment, and the effect is clear. The results also seem to verify that the polynomial approximation can achieve more flexible filters.

**Methods And Evaluation Criteria:**

The proposed approach models the graph as a system of differential equations. The coefficient matrix of this system depends entirely on the graph's properties, and then induces the corresponding solution space. Based on the solution space, the method offers a clear description of node feature changes or feature directions during message passing, significantly enhancing the interpretability of graph neural networks. Compared to traditional GNNs, it offers a more accurate depiction of graph node embedding spaces during message passing. Additionally, the incorporation of polynomial approximation enhances flexibility, reduces computational complexity, and ensures the provision of more adaptive nonlinear filters in practical applications.

**Other Comments Or Suggestions:**

Please see pros & cons. Besides, there are suggestions:
- In Table 1, the unit basis vectors in Euclidean space are typically formatted as bold italics (e.g., $\bm{e}_i$). The current writing may require further consideration.
- In Appendix A.2, the notation C in Equation (27) of Theorem A.2 denotes the equation system's coefficient matrix. This differs mathematically from the C in Equation (28), suggesting appropriate notational distinction.

**Other Strengths And Weaknesses:**

Strengths:
- This paper is mostly clear with a well-organized structure and mathematical proofs.
- The proposed method models the graph as a system of hyperbolic partial differential equations, constructing an embedding variation space for graph nodes. This effectively captures the direction of node feature changes or significant feature directions during message passing, offering enhanced scalability and interpretability.
- The designed approach incorporates polynomial approximation into the Laplacian within the framework of differential equations, achieving significantly enhanced flexibility.
- The experimental results across diverse graph datasets demonstrate the effectiveness of the proposed method. Furthermore, extensive experiments conducted on image datasets confirm its operational flexibility and practical applicability.

Weaknesses:
- The possible computational costs under different modules are not analyzed.
- The performance of the constructed GNN may vary across different hyperbolic equations.
- Although the forward Euler method is simple, the implicit method may work better.

**Questions For Authors:**

- Considering the computational cost of the different components of the analysis approach may further strengthen the paper's point of view.
- Will different polynomials affect the performance of the proposed GNN?

**Relation To Broader Scientific Literature:**

Compared to traditional spectral GNNs, this paper proposes a PDE-based model to achieve message passing mechanism. Besides, compared with the current differential equation-based paradigm, it provides better scalability and interpretability.

**Theoretical Claims:**

In this paper, Theorems 3.1 and 3.3 are the core parts to prove the existence of solutions and the structure of the solution space. For the process of proof, the key is to transform the original partial differential equations into a system of homogeneous linear ordinary differential equations with constant coefficients by variable substitution, and then utilize the solution theory of ordinary differential equations. The proof is relatively clear and rigorous.

---

> ### Author Rebuttal · Authors · 2025-04-01
>
> **[Cons 1/Q1]**: *Computational costs under different modules.*
>
> **[Answer]**: The efficiency of the proposed method primarily depends on the choice of the spectral GNN filter. A more efficient filter leads to higher efficiency. Additionally, hyperparameters ($i.e.$, termination time $T$ and time step $\tau$) also affect the efficiency of the model. A smaller $T/\tau$ results in fewer iteration steps and higher efficiency.
>
> **[Cons 2]**: *The impact of different hyperbolic equations.*
>
> **[Answer]**: In essence, this work proposes a spectral GNN framework applicable to a broad class of hyperbolic PDEs. The specific GNN architecture for particular equations can be derived following this framework. In dynamical system modeling, second-order hyperbolic equations are most commonly encountered, characterized by real-valued eigenvalues and wave-like solution behavior.
>
> The computational performance of the constructed GNN primarily depends on the numerical methods employed. The explicit scheme induced by the forward Euler method, when combined with initial conditions, enables rapid iterative solving, whereas implicit backward schemes may result in slower computation speeds. Different hyperbolic equations may exhibit variations in numerical stability. We will further investigate potential subtle distinctions arising from varying forms of hyperbolic equations in future works.
>
> **[Cons 3]**: *The implicit method may work better.*
>
> **[Answer]**: The advantage of the forward Euler method lies in its **simplicity and efficiency**. The implicit method offers better performance but involves solving linear equation systems, incurring higher costs. Literature [1] demonstrates that the performance improvement brought by the implicit method is small, yet it triggers significant performance costs.
>
> [1] Ben Chamberlain, James Rowbottom, Maria I. Gorinova, Michael M. Bronstein, Stefan Webb, Emanuele Rossi: GRAND: Graph Neural Diffusion. ICML 2021: 1407-1418.
>
> **[Q2]**: *The effect of different polynomials.*
>
> **[Answer]**: As shown in Tables 4 and 5, polynomials such as Bernstein and Jacobi can fit arbitrary filters, making them superior GNNs. Chebyshev polynomials exhibit the Runge phenomenon, leading to the fitting of suboptimal filters. Since our method enhances the capability of polynomial fitting for filters, it results in significant performance improvements for Chebyshev polynomials.
>
> **[Sug]**: *Improvement of writing.*
>
> **[Answer]**: We will consider and incorporate your suggestions into the revised version. Thanks for your comments.

---

### Official Review · Reviewer_fiaR · 2025-03-09

**Overall Recommendation:** 4

**Summary:**

This paper introduces "Hyperbolic GNNs" (Hyperbolic Graph Neural Networks), a type of spectral graph neural network where the message passing is implemented through hyperbolic partial differential equations. The network can thus be viewed as a kind of dynamical system. The authors provide several experiments demonstrating the superior performance of the hyperbolic GNNs compared to other spectral graph networks.

**Claims And Evidence:**

Yes, it is my opinion that the claims are backed up by both theoretical and experimental evidence.

**Essential References Not Discussed:**

Here comes my main criticism of this paper. The paper introduces a type of network that they refer to as "Hyperbolic GNN". But the hyperbolic graph neural networks already exist in the literature. As far as I know they were first introduced in the paper

"Hyperbolic Graph Neural Networks" by Liu, Nickel, Kiela
(Neurips 2019), arXiv:1910.12892

After reading the paper under review as well as the paper by Liu et al, I have come to the conclusion that these papers are not referring to the same kind of networks at al. As mentioned above, in the paper under review the term "hyperbolic" enters because the message passing is implemented using hyperbolic PDEs. However, in the paper by Liu et al, the term "hyperbolic" refers to the underlying hyperbolic geometry, i.e. the signature of the manifold on which the network is defined.
According to me understanding these are fundamentally different.

Moreover, there is also the paper

"Hyperbolic Graph Convolutional Neural Networks" by Chami et al
(Neurips 2019)

which introduces a similar structure for convolutional networks.

So, first of all, I think it is very strange that the authors of the present paper would introduce the name "Hyperbolic GNNs" without any type of mentioning or comparison with the previous papers on hyperbolic graph neural networks.

If indeed my understanding is correct, and these are inherently different structures, then I think it would be appropriate for the authors of the present paper to change the name of their networks, and the title of their paper. The name "hyperbolic graph neural networks" is taken and they have to come up with something else.

So until this issue has been resolved I cannot recommend publication of the present paper.

Update after rebuttal: I approve of the authors changes and I will modify my overall assessment accordingly.

**Experimental Designs Or Analyses:**

No

**Methods And Evaluation Criteria:**

Yes

**Other Comments Or Suggestions:**

See above

**Other Strengths And Weaknesses:**

See my comments above.

**Questions For Authors:**

NA

**Relation To Broader Scientific Literature:**

See below

**Theoretical Claims:**

I have checked parts of the proofs of the Theorems and they seem logically sound and correct to me.

---

> ### Author Rebuttal · Authors · 2025-04-01
>
> **[Cons]**: *The question about the concept of hyperbolic PDE.*
>
> **[Answer]**: We appreciate your theoretical endorsement for our approach. As you pointed out, the hyperbolic PDE in this paper differs fundamentally from the hyperbolic geometry mentioned in the literature [1,2].
>
> The focus of hyperbolic PDEs lies in constructing the **message passing process** of GNNs ($i.e.$, $\frac{\partial^2 \mathbf{X}}{\partial t^2} = a^2 \widehat{\mathbf{L}} \mathbf{X}$) based on the theory of partial differential equations. This design can naturally map nodes into a solution space spanned by a set of eigenvectors of the graph, which improves the capability of learning complex graph filters, such as handling the challenging heterophilic graphs.
>
> On the other hand, hyperbolic geometry focuses more on the **hierarchical structure** of graphs. It typically involves mapping nodes to hyperbolic space using models such as the Poincaré Ball Model ($i.e.$, $d(\mathbf{x}, \mathbf{y})=\mathrm{arcosh}(1+2\frac{||\mathbf{x}-\mathbf{y}||^2}{(1-||\mathbf{x}||^2)(1-||\mathbf{y}||^2)})$) or the Lorentz Model ($i.e.$, $<\mathbf{x}, \mathbf{y}>=-x_0y_0+\sum^n_{i=1}x_ny_n$) to extract hierarchical relationships between nodes.
>
> The essence of the two concepts mentioned above is fundamentally different, with their focus being distinct. To avoid ambiguity for the readers, we will incorporate your valuable suggestions and modify the title as well as the method names in the next version. We will also discuss the distinctions from hyperbolic geometry in related works to avoiding misunderstanding.
>
> Below are our new title and method name for modification:
> ***"Hyperbolic-PDE GNN: Spectral Graph Neural Networks in the Perspective of A System of Hyperbolic Partial Differential Equations"***
>
> [1] Qi Liu, Maximilian Nickel, Douwe Kiela: Hyperbolic Graph Neural Networks. NeurIPS 2019.
> [2] Ines Chami, Zhitao Ying, Christopher Ré, Jure Leskovec: Hyperbolic Graph Convolutional Neural Networks. NeurIPS 2019.

---

> > ### Comment · Reviewer_fiaR · 2025-04-02
> >
> > Thanks for your rebuttal. I approve of your changes and will update my review accordingly.

---

### Official Review · Reviewer_TsKB · 2025-03-10

**Overall Recommendation:** 4

**Summary:**

This paper proposes a Hyperbolic Graph Neural Network (GNN) framework based on a system of hyperbolic partial differential equations (PDEs), establishing a novel message-passing paradigm that derives topology-aware node representations by solving these equations. Supported by a solid theoretical foundation and comprehensive empirical validation, the method demonstrates both effectiveness and flexibility. This work makes a significant contribution to redefining GNN architectures and advancing the field.

Thanks to the author for the reply, this solved most of my problems, I will keep my score.

**Claims And Evidence:**

The authors provide thorough theoretical proofs and experimental evidence to support their claims.

**Essential References Not Discussed:**

N/A

**Experimental Designs Or Analyses:**

The experiments on node classification and image filtering tasks are well-designed and adhere to established practices in spectral GNN research. The selection of datasets, baselines, and evaluation protocols is appropriate, ensuring reliability and reproducibility.

**Methods And Evaluation Criteria:**

Extensive experiments on node classification and image signal filtering tasks validate the effectiveness and robustness.

**Other Comments Or Suggestions:**

The theoretical derivations (e.g., Appendices A.2–A.3) enhance interdisciplinary accessibility. While the framework excels in spectral GNNs, its applicability to non-spectral architectures (e.g., spatial GNNs) warrants further exploration.

**Other Strengths And Weaknesses:**

Strengths:
1.	Novelty: The framework introduces a principled connection between node representations and topology via hyperbolic PDEs, providing a theoretical and novel message-passing paradigm.
2.	Theoretical Interpretability: The derivation of solution spaces and their decomposition into topology-dependent components is mathematically sound and clearly articulated.
3.	Significance: Beyond proposing a practical GNN, the work provides theoretical insights into PDE-based methods, facilitating deeper understanding of related approaches.

Weaknesses:
1.	Some of results on the node classification task remains slightly below state-of-the-art benchmarks (in Table 3).
2.	A limited discussion of complexity and parameter efficiency may hinder practical performance. It would be better to conduct more analysis.

**Questions For Authors:**

1.	What are the practical and theoretical distinctions between different polynomial families (e.g., Chebyshev vs. Bernstein) in approximating the solution space? Which family yields optimal empirical results?
2.	Could the hyperbolic framework be adapted to non-spectral GNNs (e.g., attention-based models)? What challenges might arise?
3.	The framework introduces additional hyperparameters (e.g., polynomial order). How should practitioners balance expressivity and computational overhead?

**Relation To Broader Scientific Literature:**

This work leverages mathematical-physical methods to address AI challenges. While existing graph studies often employ diffusion equations empirically, this paper particularly indicates the effectiveness of PDE-based approaches through topology-aware solution spaces, providing theoretical improvements to existing works.

**Theoretical Claims:**

The appendix includes informative theoretical proofs. The existence of solutions to the hyperbolic PDE system is formally established (Appendix A.2), and the simplified solution space linking node features to topology is derived and analyzed (Appendix A.3). The mathematical foundations are robust and align with established principles.

---

> ### Author Rebuttal · Authors · 2025-04-01
>
> **[Cons 1]**: *Slightly lower performance on some datasets.*
>
> **[Answer]**: In this paper, we aim to propose a general framework for GNNs, with the advantage of being applicable to spectral GNNs. The hyperbolic PDE-based paradigm allows the model to learn complex graph filters, generating better results on heterophilic graphs. Extensive results demonstrate the superiority, and reflect a new research direction of GNNs. We leave further improvements in future works.
>
> **[Cons 2/Q3]**: *Complexity and parameter efficiency.*
>
> **[Answer]**: Converting spectral GNNs into a dynamical system may increase the the computational costs, but can also be controlled by hyperparameters (termination time $T$ and time step $\tau$). For instance, when $T=1$ and $\tau=1$, the efficiency is comparable to the original method while performing better. Furthermore, this method introduces no additional parameters apart from the MLP parameters for initializing $\varphi(\cdot)$. Therefore, the cost and efficiency is acceptable and worthwhile.
>
> **[Q1]**: *The theoretical distinctions between different polynomial families.*
>
> **[Answer]**: Different polynomials, such as Chebyshev and Bernstein polynomials, would fit filters **based on their mathematical properties**. For example, Chebyshev polynomials exhibit Runge's phenomenon, while Bernstein polynomials can fit any function. Experimental results indicate that in graph tasks, learnable GPR performs the best, whereas ChebNetII may excel in image tasks.
>
> **[Q2]**: *Applicability of hyperbolic framework on non-spectral GNNs.*
>
> **[Answer]**: Our method is applicable to non-spectral GNNs. We can simply replace the function $P(\cdot)$ in Equation 17 with any feasible convolution operation such as attention mechanisms, as demonstrated in our enhancement of GAT on image tasks. The effectiveness of this framework depends on whether the base model can effectively adapt to the task. Attention mechanisms can effectively capture the importance levels between pixels, thus Hyperbolic-GAT further improves the performance of GAT.

---

### Official Review · Reviewer_1BNk · 2025-03-13

**Overall Recommendation:** 3

**Summary:**

This paper proposes to formulate message passing in spectral GNNs as a system of hyperbolic PDEs by extending the concepts of
gradient and divergence on manifolds to graphs. Based on this formulation, node features are shown to propagate messages along specific directions of eigenvectors and therefore better capture the topology of graphs. To improve the efficiency of the model, polynomials are used to approximate the solution. Experimental results on some graph tasks demonstrate the effectiveness of the proposed method.

**Claims And Evidence:**

Yes

**Essential References Not Discussed:**

No

**Experimental Designs Or Analyses:**

Yes

**Methods And Evaluation Criteria:**

Yes

**Other Comments Or Suggestions:**

Please see the questions.

**Other Strengths And Weaknesses:**

Strengs:
- The exposition is smooth and easy to follow
- The proposed idea of formulating message passing as a system of hyperbolic PDEs seems novel
- Various spectral GNNs are considered to demonstrate the practical benefits of the proposed method


Weaknesses:
- Experimental evaluation for a variety of graph tasks is somewhat limited
- The proposed method only shows marginal improvements over state-of-the-art methods on the node classification task
- The mathematical proof of Theorem 3.5 is not provided

**Questions For Authors:**

- The proof of Theorem 3.5 is not provided or I am missing something ?
- Why other popular graph learning tasks like graph classification or link prediction are not considered ?
- Since the message passing scheme is formulated as a system of hyperbolic PDEs, I am wondering if the proposed model is advantagous over existing ones on graph datasets with strong hierarchical structures (e.g., Disease, Airport) ? If that is not the case, could the authors elaborate ?

**Relation To Broader Scientific Literature:**

The proposed idea seems novel. It generally improves existing spectral GNNs and the interpretability of message passing scheme

**Theoretical Claims:**

I read all the proofs to get the general ideas but could not follow all the details

---

> ### Author Rebuttal · Authors · 2025-04-01
>
> **[Cons 1/2]**: *Somewhat limited experimental evaluation, and some marginal improvements on a few datasets.*
>
> **[Answer]**: The proposed method is essentially a general framework that enhances the capability of spectral GNNs, as shown in Tables 4 and 5. The results on both homophilic and heterophilic graphs, and the extensive experimets on the low-pass filter and arbitrary filter methods, consistently achieve competitive results. All the analyses comprehensively demonstrate the effectiveness of our method.
>
> On the other hand, our method also demonstrates high interpretability to connect spectral GNNs. It provide high flexibility to implement different spectral graph filters for further improvements, and reflect a new direction of GNN studies.
>
> Thanks for the comments. We will conduct further exploration in our future works.
>
> **[Cons 3/Q1]**: The lack of the proof of Theorem 3.5.
>
> **[Answer]**: Theorem 3.5 is a well-known mathematical theory as Weierstrass approximation theorem [1,2], and we will introduce more details and references in the next version.
>
> [1] Stone, M. H. (1937), "Applications of the Theory of Boolean Rings to General Topology", Transactions of the American Mathematical Society, 41 (3): 375–481, doi:10.2307/1989788, JSTOR 1989788.
>
> [2] https://en.wikipedia.org/wiki/Stone%E2%80%93Weierstrass_theorem
>
> **[Q2]**: *Evaluation without graph classification or link prediction.*
>
> **[Answer]**: Actually, our method is a foundational and general framework, which is a **task-agnostic** graph learning method. The node classification task is a widely-adopted evaluation setting for evaluating the effectivness of GNNs [1,2,3], and thus we follow this setting. The extensive results also reflect the superiority.
> Thanks for the suggestion, we would like conduct more explorations on link prediction and graph classification in our future works.
>
> [1] Wang, X. and Zhang, M. How powerful are spectral graph neural networks. ICML 2022.
> [2] Geng, H., Chen, C., He, Y., Zeng, G., Han, Z., Chai, H., and Yan, J. Pyramid graph neural network: A graph sampling and filtering approach for multi-scale disentangled representations. KDD 2023.
> [3] Zheng, S., Zhu, Z., Liu, Z., Li, Y., and Zhao, Y. Node-oriented spectral filtering for graph neural networks. IEEE Trans. Pattern Anal. Mach. Intell., 46(1):388–402.
>
> **[Q3]**: *Comparison with methods for hierarchical structures.*
>
> **[Answer]**: Actually, our hyperbolic PDE-based GNNs and traditional hyperbolic GNNs are quite different paradigms.
> - Traditional hyperbolic GNNs map nodes to hyperbolic embedding space, which can be capable to capture hierarchical structures.
> - This paper proposes a new paradigm (Equation 9 and 24) of message passing through **a system of hyperbolic PDEs**. This paradigm can naturally map nodes into a solution space spanned by a set of eigenvectors of the graph, which improves the capability of learning complex graph filters, such as handling the challenging heterophilic graphs.
>
> Therefore, the motivations, modeling paradigms and properties are different. Please refer to #R3 for more details. Thanks for your suggestions. We will make more discussion in the next version.

---

> > ### Comment · Reviewer_1BNk · 2025-04-08
> >
> > I would like to thank the authors for their clarification that addresses my concerns. I will keep my initial score.

---

### Decision · Program_Chairs · 2025-05-01

**Decision:**

Accept (poster)

**Comment:**

This paper introduces Hyperbolic GNNs, a type of spectral graph neural network where the message passing is implemented through hyperbolic partial differential equations. Several theoretical advantages of hyperbolic dynamic systems are highlighted to be beneficial in the scope of GNNs. Some nice experimental results are also given to support the proposed method. Here, all reviewers support the publication of the work. In the discussion period, the reviewers agree that the method is not entirely novel, as hyperbolic GNNs were already suggested in the papers (Eliasof et al., 2021) and (Rusch et al., 2022). In light of the positive reviews, and after discussion, I vote for the acceptance of the paper. In the camera-ready version, the authors should revise their paper to give (Eliasof et al., 2021) and (Rusch et al., 2022) credit for being the first to propose hyperbolic-type GNN architectures.